# `POLY-HOOT`: Monte-Carlo Planning in Continuous Space MDPs with Non-Asymptotic Analysis

**Weichao Mao**
ECE and CSL
University of Illinois at Urbana-Champaign
`weichao2@illinois.edu`

**Kaiqing Zhang**
ECE and CSL
University of Illinois at Urbana-Champaign
`kzhang66@illinois.edu`

**Qiaomin Xie**
ORIE
Cornell University
`qiaomin.xie@cornell.edu`

**Tamer Başar**
ECE and CSL
University of Illinois at Urbana-Champaign
`basar1@illinois.edu`

## Abstract

Monte-Carlo planning, as exemplified by Monte-Carlo Tree Search (MCTS), has demonstrated remarkable performance in applications with finite spaces. In this paper, we consider Monte-Carlo planning in an environment with continuous state-action spaces, a much less understood problem with important applications in control and robotics. We introduce `POLY-HOOT`, an algorithm that augments MCTS with a continuous armed bandit strategy named Hierarchical Optimistic Optimization (HOO) (Bubeck et al., 2011). Specifically, we enhance HOO by using an appropriate *polynomial*, rather than *logarithmic*, bonus term in the upper confidence bounds. Such a polynomial bonus is motivated by its empirical successes in AlphaGo Zero (Silver et al., 2017b), as well as its significant role in achieving theoretical guarantees of finite space MCTS (Shah et al., 2019). We investigate, for the first time, the regret of the enhanced HOO algorithm in non-stationary bandit problems. Using this result as a building block, we establish non-asymptotic convergence guarantees for `POLY-HOOT`: the value estimate converges to an arbitrarily small neighborhood of the optimal value function at a polynomial rate. We further provide experimental results that corroborate our theoretical findings.

## 1 Introduction

Monte-Carlo tree search (MCTS) has recently demonstrated remarkable success in deterministic games, especially in the game of Go (Silver et al., 2017b), Chess and Shogi (Silver et al., 2017a). It is also among the very few viable approaches to problems with partial observability, e.g., Poker (Rubin and Watson, 2011), and problems involving highly complicated strategies like real-time strategy games (Uriarte and Ontanón, 2014). However, most Monte-Carlo planning solutions only work well in finite state and action spaces, and are generally not compatible with continuous action spaces with enormous branching factors. Many important applications such as robotics and control require planning in a continuous state-action space, for which feasible solutions, especially those with theoretical guarantees, are scarce. In this paper, we aim to develop an MCTS method for *continuous* domains with *non-asymptotic convergence* guarantees.

Rigorous analysis of MCTS is highly non-trivial even in finite spaces. One crucial difficulty stems from the fact that the state-action value estimates in MCTS are non-stationary over multiple simulations, because the policies in the lower levels of the search tree are constantly changing. Due to the strong non-stationarity and interdependency of rewards, the reward concentration hypothesis made

in the seminal work of Kocsis and Szepesvári (2006)—which provides one of the first theoretical analysis of bandit-based MCTS—turns out to be unrealistic. Hence, the convergence analysis given in Kocsis and Szepesvári (2006) is unlikely to hold in general. Recently a rigorous convergence result is established in Shah et al. (2019), based on further investigation of *non-stationary multi-armed bandits* (MABs).

Besides the non-stationarity issue inherent in MCTS analysis, an additional challenge for continuous domains lies in balancing the trade-off between generating fine-grained samples across the entire continuous action domain to ensure optimality, and guaranteeing sufficient exploitation of the sampled actions for accurate estimations. To tackle this challenge, a natural idea is to manually discretize the action space and then solve the resulting discrete problem using a discrete-space planning algorithm. However, this approach inevitably requires a hyper-parameter pre-specifying the level of discretization, which in turn leads to a fundamental trade-off between the computational complexity and the optimality of the planning solution: coarse discretization often fails to identify the optimal continuous action, yet fine-grained discretization leads to a large action space and heavy computation.

In this paper, we consider Monte-Carlo planning in continuous space Markov Decision Processes (MDPs) without manually discretizing the action space. Our algorithm integrates MCTS with a continuous-armed bandit strategy, namely Hierarchical Optimistic Optimization (HOO) (Bubeck et al., 2011). Our algorithm *adaptively partitions* the action space and quickly identifies the region of potentially optimal actions in the continuous space, which alleviates the inherent difficulties encountered by pre-specified discretization. The integration of MCTS with HOO has been empirically evaluated in Mansley et al. (2011), under the name of the Hierarchical Optimistic Optimization applied to Trees (HOOT) algorithm. HOOT directly replaces the UCB1 bandit algorithm (Auer et al., 2002) used in finite-space MCTS with the HOO strategy. However, this algorithm has a similar issue as that in Kocsis and Szepesvári (2006), as they both use a *logarithmic* bonus term for bandit exploration instead of a *polynomial* term. As pointed out in Shah et al. (2019) and mentioned above, convergence guarantees of these algorithms are generally unclear due to the lack of concentration of non-stationary rewards. In this work, we enhance the HOO strategy with a polynomial bonus term to account for the non-stationarity. As we will show in our theoretical results, our algorithm, Polynomial Hierarchical Optimistic Optimization applied to Trees (`POLY-HOOT`), provably converges to an arbitrarily small neighborhood of the optimum at a polynomial rate.

**Contributions.** First, we enhance the continuous-armed bandit strategy HOO, and analyze its regret concentration rate in a non-stationary setting, which may also be of independent theoretical interest in the context of bandit problems. Second, we build on the enhanced HOO to design a Monte-Carlo planning algorithm `POLY-HOOT` for solving continuous space MDPs. Third, we generalize the recent analytical framework developed for finite-space MCTS (Shah et al., 2019) and prove that the value estimate of `POLY-HOOT` converges to an arbitrarily small neighborhood of the optimal value function at a polynomial rate. We note that HOOT is among the very few MCTS algorithms for continuous spaces and popular in practice. `POLY-HOOT` improves upon HOOT and provides theoretical justifications thereof. Finally, we present experimental results which corroborate our theoretical findings and demonstrate the superior performance of `POLY-HOOT`.

**Related Work.** One of the most popular MCTS methods is the Upper Confidence Bounds applied to Trees (UCT) algorithm (Kocsis and Szepesvári, 2006), which applies the UCB1 (Auer et al., 2002) bandit algorithm for action selection. A convergence result of UCT is provided in Kocsis and Szepesvári (2006). However, this result relies on the assumption that bandit regrets under UCB1 concentrate exponentially, which is unlikely to hold in general. Recent work in Shah et al. (2019) provides a complete analysis of UCT through a further study of non-stationary bandit algorithms using polynomial bonus. Our analysis falls into the general framework proposed therein. We note that many variations and enhancements of MCTS have been developed (Coquelin and Munos, 2007; Schadd et al., 2008; Kaufmann and Koolen, 2017; Xiao et al., 2019; Jonsson et al., 2020); we refer interested readers to a survey by Browne et al. (2012). We remark that most variants are restricted to finite-action problems.

MCTS for continuous-space MDPs has been relatively less studied. In the literature a progressive widening (PW) technique (Chaslot et al., 2007; Auger et al., 2013) is often used to discretize the action space and ensure sufficient exploitation. However, PW mainly concerns *when* to sample a new action, but not *how*. For example, Auger et al. (2013) draws an action uniformly at random, which is sample-inefficient compared to our bandit-based action selection. Popular in empirical work is

the HOOT algorithm in (Mansley et al., 2011), which directly replaces the UCB1 bandit strategy in UCT with HOO. This work does not provide theoretical guarantees, and given the non-stationarity of the bandit rewards, there is a good reason to believe that a more sophisticated variant of HOO is needed. An open-loop planning solution named Hierarchical Open-Loop Optimistic Planning (HOLOP) is proposed and empirically evaluated in Weinstein and Littman (2012). In Yee et al. (2016), MCTS is combined with kernel regression, and the resulting algorithm demonstrates good empirical performance. More recently, Kim et al. (2020) proposes to partition the continuous space based on the Voronoi graph, but they focus on deterministic rewards and do not utilize bandits to *guide the exploration and exploitation* of actions, which is the main focus of our work.

**Outline.** The rest of the paper is organized as follows: In Section 2, we introduce the mathematical formulation and some preliminaries. In Section 3, we present our `POLY-HOOT` algorithm. In Section 4, we provide our analysis of the non-stationary bandits and our main results on the convergence of `POLY-HOOT`. Simulation results are provided in Section 5. Finally, we conclude our paper in Section 6. The detailed algorithms and proofs of the theorems can be found in the appendix.

## 2  Preliminaries

### 2.1  Markov Decision Processes

We consider an infinite-horizon discounted MDP defined by a 5-tuple $(S, A, T, R, \gamma)$, where $S \subseteq \mathbb{R}^n$ is the continuous state space, $A \subseteq \mathbb{R}^m$ the continuous action space, $T : S \times A \to S$ the deterministic transition function, $R : S \times A \to [-R_{max}, R_{max}]$ the (bounded) stochastic reward function, and $\gamma \in (0, 1)$ is the discount factor. We do not require $S$ and $A$ to be compact, thus our theory covers many control applications with possibly unbounded state-action spaces. The assumption of deterministic state transitions is common in the MCTS literature (Browne et al., 2012; Shah et al., 2019; Kim et al., 2020), as MCTS was historically introduced and popularly utilized in problems like Go (Gelly et al., 2006; Silver et al., 2017b) and Atari games (Guo et al., 2014). For simplicity we use the notation $s \circ a \triangleq T(s, a)$ to denote the next state deterministically reached by taking action $a \in A$ at the current state $s \in S$.

A policy $\pi : S \to A$ specifies the action $a = \pi(s)$ taken at state $s$. The value function $V^\pi : S \to \mathbb{R}$ of a policy $\pi$ is defined as the expected discounted sum of rewards following $\pi$ starting from the current state $s \in S$, i.e., $V^\pi(s) = \mathbb{E}_\pi \left[ \sum_{t=0}^\infty \gamma^t R(s_t, a_t) | s_0 = s \right]$. Similarly, define the state-action value function $Q^\pi(s, a) = \mathbb{E}_\pi \left[ \sum_{t=0}^\infty \gamma^t R(s_t, a_t) | s_0 = s, a_0 = a \right]$. The planner aims to find an optimal policy $\pi^*$ that achieves the maximum value $V^{\pi^*}(s) = V^*(s) \triangleq \sup_\pi V^\pi(s)$ for all $s \in S$.

We consider the problem of computing the optimal value function for any given input state, with access to a generative model (or simulator) of the MDP. A generative model provides a randomly sampled next state and reward, when given any state-action pair $(s, a)$ as input. Our algorithms and results readily extend to learning the optimal policy or Q-function.

### 2.2  Monte-Carlo Tree Search

To estimate the optimal value of a given state, Monte-Carlo tree search (MCTS) builds a multi-step look-ahead tree, with the state of interest as the root node, using Monte-Carlo simulations (Browne et al., 2012). Each node in the tree represents a state, and each edge represents a state-action pair that leads to a child node denoting the subsequent state. At each iteration, starting from the root node, the algorithm selects actions according to a *tree policy* and obtains samples from the generative model until reaching a leaf node. An estimate for the value of leaf node can be either obtained by simulations of a roll-out policy or given by some function approximation. The leaf node estimate and samples generated along the path are then backed-up to update the statistics of selected nodes. The tree policy plays a key role of balancing exploration-exploitation. The most popular tree policy is UCT (Kocsis and Szepesvári, 2006), which selects children (actions) according to the Upper Confidence Bound (UCB1) (Auer et al., 2002) bandit algorithm. Note that UCT, and most variants thereof, are restricted to the finite action setting.

A major challenge in the theoretical analysis of any MCTS algorithm is the *non-stationarity* of bandit rewards. Specifically, since the policies at the lower level bandits of MCTS are constantly changing, the reward sequences for each bandit agent drift over time, causing the reward distribution

to be highly non-stationary. The performance of each bandit depends on the results of a chain of bandits at the lower levels, and this hierarchical inter-dependence of bandits makes the analysis highly non-trivial. A complete solution to address this non-stationarity has been given recently in Shah et al. (2019), where the authors inductively show the polynomial concentration of rewards by leveraging a non-stationary bandit algorithm with a *polynomial* bonus term. Our approach in the continuous case is based upon a similar reasoning as in Shah et al. (2019).

## 2.3 Hierarchical Optimistic Optimization

HOO (Bubeck et al., 2011) is an extension of finite-armed bandit algorithms to problems with arms living in an arbitrary measurable space, e.g., the Euclidean space. HOO incrementally builds a binary tree covering of the continuous action space $X$. Each node in the tree covers a subset of $X$. This subset is further divided into two, corresponding to the two child nodes. HOO selects an action by following a path from the root node to a leaf, and at each node it picks the child node that has the larger upper confidence bound (to be precise, larger $B$-value; see equation (2)) for the reward. In this manner, HOO adaptively subdivides the action space and quickly focuses on the area where potentially optimal actions lie in.

Following the notations in Bubeck et al. (2011), we index the nodes in the above HOO tree by pairs of integers $(h, i)$,[1] where $h \geq 0$ denotes the depth of the node, and $1 \leq i \leq 2^h$ denotes its index on depth $h$. In particular, the root node is $(0, 1)$; the two children of $(h, i)$ are $(h + 1, 2i - 1)$ and $(h + 1, 2i)$. Let $\mathcal{P}_{h,i} \subseteq X$ be the domain covered by the node $(h, i)$. By definition, we have $\mathcal{P}_{0,1} = X$ and $\mathcal{P}_{h,i} = \mathcal{P}_{h+1,2i-1} \cup \mathcal{P}_{h+1,2i}, \forall h \geq 0$ and $1 \leq i \leq 2^h$. Let $\mathcal{C}(h, i)$ denote the set of all descendants of node $(h, i)$. Let $(H_t, I_t)$ denote the node played by HOO at round $t$, with observed reward $Y_t$. Then the number of times that a descendant of $(h, i)$ has been played up to and including round $n$ is denoted by $T_{h,i}(n) = \sum_{t=1}^{n} \mathbb{1}_{\{(H_t, I_t) \in \mathcal{C}(h,i)\}}$, and the empirical average of rewards is defined as $\widehat{\mu}_{h,i}(n) = \frac{1}{T_{h,i}(n)} \sum_{t=1}^{n} Y_t \mathbb{1}_{\{(H_t, I_t) \in \mathcal{C}(h,i)\}}$.

In the original HOO algorithm of Bubeck et al. (2011), the upper confidence bound of a node $(h, i)$ is constructed using a logarithmic bonus term:

$$U_{h,i}(n) = \begin{cases} \widehat{\mu}_{h,i}(n) + \sqrt{\frac{2 \ln n}{T_{h,i}(n)}} + \nu_1 \rho^h, & \text{if } T_{h,i}(n) > 0, \\ \infty, & \text{otherwise}, \end{cases} \tag{1}$$

where $\nu_1$ and $\rho$ are two constants that characterize the reward function and the action domain. Given $U_{h,i}(n)$, one further introduces a critical quantity termed the $B$-values:

$$B_{h,i}(n) = \begin{cases} \min \{U_{h,i}(n), \max \{B_{h+1,2i-1}(n), B_{h+1,2i}(n)\}\}, & \text{if } (h, i) \in \mathcal{T}_n, \\ \infty, & \text{otherwise}, \end{cases} \tag{2}$$

where $\mathcal{T}_n$ is the set of nodes that are already included in the binary tree at round $n$. Starting from the root node, HOO iteratively selects a child node with a larger $B$-value until it reaches a leaf node, which corresponds to an arm of the bandit to be pulled.

## 3 Algorithm: `POLY-HOOT`

Our algorithm for continuous space MCTS, Polynomial Hierarchical Optimistic Optimization applied to Trees (`POLY-HOOT`), is presented in Algorithm 1.

`POLY-HOOT` follows a similar framework as the classic UCT algorithm, but has the following critical enhancements to handle continuous spaces with provable convergence guarantees.

**1. HOO-Based Action Selection.** We replace the discrete UCB1 bandit agent with a continuous-armed HOO agent. In this case, each node in the Monte-Carlo tree is itself a HOO tree. In particular, `POLY-HOOT` invokes the HOO algorithm through two functions: the $HOO\_query$ function selects actions; after the action is taken and the reward is realized, the $HOO\_update$ function updates the reward information at each HOO agent along the Monte-Carlo sampling path. Detailed descriptions are provided in Appendix A.

**Algorithm 1:** `POLY-HOOT`

---

1  **Input:** value oracle at leaf nodes $\hat{V}$, root node $s^{(0)}$, maximum search depth $D$, number of MCTS
    simulations $n$, and parameters $\{\alpha^{(i)}\}_{i=0}^{D-1}, \{\xi^{(i)}\}_{i=0}^{D-1}, \{\eta^{(i)}\}_{i=0}^{D-1}$.

2  **Output:** value estimate of the root node $s^{(0)}$.

3  **for** *simulation round* $t \leftarrow 1$ *to* $n$ **do**

4      **for** *depth* $d \leftarrow 0$ *to* $D-1$ **do**

5          $a^{(d)} \leftarrow HOO\_query(d, s^{(d)}, t)$ with depth limitation $\bar{H}$;

6          $r^{(d)} \sim R(s^{(d)}, a^{(d)})$;

7          $s^{(d+1)} \leftarrow s^{(d)} \circ a^{(d)}$;

8      $r^{(D)}(s^{(D)}) \leftarrow \hat{V}(s^{(D)})$;

9      **for** *depth* $d \leftarrow 0$ *to* $D-1$ **do**

10         $Y^{(d)} \leftarrow r^{(d)} + \gamma r^{(d+1)} + \cdots + \gamma^{D-d-1} r^{(D-1)} + \gamma^{D-d} r^{(D)}(s^{(D)})$;

11         $\tilde{v}^{(d)}(s^{(d)}) \leftarrow \tilde{v}^{(d)}(s^{(d)}) + Y^{(d)}$;

12         $HOO\_update(d, s^{(d)}, t, Y^{(d)})$ using parameters $\alpha^{(d)}, \xi^{(d)}$ and $\eta^{(d)}$;

13 **return** $\tilde{v}^{(0)}(s^{(0)})/n$.

---

**2. Polynomial Bonus.** We replace the logarithmic bonus term used in the original HOO algorithm (Equation (1)) with a polynomial term. In particular, our algorithm constructs the upper confidence bound of a node $(h, i)$ as follows:

$$U_{h,i}(n) = \begin{cases} \widehat{\mu}_{h,i}(n) + n^{\alpha^{(d)}/\xi^{(d)}} T_{h,i}(n)^{\eta^{(d)}-1} + \nu_1 \rho^h, & \text{if } T_{h,i}(n) > 0, \\ \infty, & \text{otherwise}, \end{cases}$$

where $\alpha^{(d)}, \xi^{(d)}$ and $\eta^{(d)}$ are constants to be specified later for each depth $d$ in MCTS. As shall become clear in the analysis, this polynomial bonus is critical in establishing convergence of MCTS. In particular, MCTS involves a hierarchy of bandits with non-stationary rewards, for which logarithmic bonus is no longer appropriate and does not guarantee (even asymptotic) convergence. Interestingly, the empirically successful AlphaGo Zero also uses polynomial bonus (Silver et al., 2017b). As in the original HOO, our algorithm navigates down the HOO tree using the $B$-value defined in (2), except that we plug in the above polynomial upper confidence bound $U_{h,i}(n)$.

**3. Bounded-Depth HOO Tree.** We place an upper bound $\bar{H}$ on the maximum depth of the HOO tree. Every time we reach a node at the maximum depth, the algorithm repeats the action taken previously at that node. As such, our enhanced HOO stops exploring new actions after trying sufficiently many actions. In the original HOO strategy, the tree is allowed to extend infinitely deep, so that the action space can be discretized into arbitrarily fine granularity. When the bandit rewards are non-stationary, as in MCTS, this strategy might overlook the long-term optimal action and get stuck in a suboptimal area in the early stage of the tree search. On the contrary, our bounded depth HOO tree ensures that the actions already explored will be fully exploited against the non-stationarity of rewards. Our analysis shows that as long as the total number of actions tried is sufficiently large (i.e., $\bar{H}$ is chosen large enough), our algorithm still converges to an arbitrarily small neighborhood of the optimal value.

### 3.1 Analysis Setup

Setting the stage for our theoretical analysis, we introduce several useful notations. For each HOO agent, let $X \subseteq A \subseteq [0,1]^m$ denote the continuous set of actions (i.e., arms) available at the current state. Each arm $x \in X$ is associated with a stochastic payoff distribution, which corresponds to the "cost-to-go" or $Q$-value of taking action $x$ at the current state of the MDP. The expectation of this reward function at time $t$ is denoted by $f_t(x) : X \to \mathbb{R}$, which is also termed the temporary mean-payoff function at time $t$. Note that in MCTS the temporary mean-payoff functions are non-stationary over time because the cost-to-go of an action depends on the actions to be chosen later in the lower levels of MCTS. Let $f$ be the limit of $f_t$ in the sense that $f_t$ converges to $f$ in $L^\infty$ at a polynomial rate: $\|f_t - f\|_\infty \leq \frac{C}{t^\zeta}$, $\forall t \geq 1$ for some constant $C > 0$ and $\zeta \in (0, \frac{1}{2})$. The precise definition of $f_t$ and $f$, as well as the convergence of $f_t$, is formally established in Theorem 2. We call $f$ the limiting mean-payoff function (or simply the mean-payoff function).

Since the rewards of the MDP are bounded by $R_{max}$, the bandit payoff for each node at depth $d$ is bounded by $\frac{R_{max}}{1-\gamma}$, and so is the limiting mean-payoff $f$ function. Let $f^* = \sup_{x \in X} f(x)$ denote the optimal payoff at an HOO agent, and the random variable $X_t$ denote the arm selected by the agent at round $t$. The agent aims to minimize the regret in the first $n$ rounds: $R_n \triangleq nf^* - \sum_{t=1}^{n} Y_t$, where $Y_t$ is the observed payoff of pulling arm $X_t$ at round $t$, with $\mathbb{E}[Y_t] = f_t(X_t)$.

We state two assumptions that will be utilized throughout our analysis. These two assumptions are similar to Assumptions A1 and A2 in Bubeck et al. (2011). For each HOO agent in MCTS, given the parameters $\nu_1$ and $\rho \in (0, 1)$, and the tree of coverings $(\mathcal{P}_{h,i})$, we assume that there exists a dissimilarity function $\ell : X \times X \to [0, \infty]$ such that the following holds.

**Assumption 1.** *There exists a constant $\nu_2 > 0$, such that for all integers $h \geq 0$,*

(a) *$diam(\mathcal{P}_{h,i}) \leq \nu_1 \rho^h, \forall 1 \leq i \leq 2^h$, where $diam(A) \triangleq \sup_{x,y \in A} \ell(x, y)$;*

(b) *there exists an $x_{h,i}^\circ \in \mathcal{P}_{h,i}$, such that $\mathcal{B}_{h,i} \triangleq \mathcal{B}\left(x_{h,i}^\circ, \nu_2 \rho^h\right) \subset \mathcal{P}_{h,i}, \forall 1 \leq i \leq 2^h$, where $\mathcal{B}(x, \varepsilon) \triangleq \{y \in X : \ell(x, y) < \varepsilon\}$ denotes an open ball centered at $x$ with radius $\varepsilon$;*

(c) *$\mathcal{B}_{h,i} \cap \mathcal{B}_{h,j} = \emptyset$ for all $1 \leq i < j \leq 2^h$.*

*Remark* 1. Assumption 1 ensures that the diameter of $\mathcal{P}_{h,i}$ shrinks at a geometric rate as $h$ grows. This is a mild assumption, which holds automatically in, e.g., compact Euclidean spaces. In particular, if the action space is a hyperrectangle, then Assumption 1 is satisfied by setting the dissimilarity function $\ell$ to be some positive power of the Euclidean norm. For example, suppose that the action space is $[0, 1]^2$. The tree covering can be generated by cutting the hyperrectangle of $\mathcal{P}_{h,i}$ at the midpoint of its longest side (ties broken arbitrarily) to obtain $\mathcal{P}_{h+1,2i-1}$ and $\mathcal{P}_{h+1,2i}$. Assumption 1 is satisfied with $\ell$ being the Euclidean norm and the parameters $\rho = \frac{1}{2}, \nu_1 = 8$, and $\nu_2 = \frac{1}{4}$. The general form of Assumption 1 allows more flexibility in the choice of $\ell$.

**Assumption 2** (Smoothness). *The limiting mean-payoff function satisfies:*

$$f^* - f(y) \leq f^* - f(x) + \max\{f^* - f(x), \ell(x, y)\}, \quad \forall x, y \in X.$$

*Remark* 2. Assumption 2 requires some smoothness of the mean-payoff function, and is milder than the common Lipschitz continuity assumption $|f(x) - f(y)| \leq \ell(x, y), \forall x, y \in X$. In particular, it requires Lipschitz continuity only in the neighborhood of any global optimal arm $x^*$, and imposes a weaker constraint for other $x \in X$. In the context of MDPs, this assumption stipulates that the $Q(s, a)$ function, after $d \in [1, D)$ steps of value iterations starting from $\hat{V}$, is a Lipschitz continuous function of the action $a$. Assumption 2 is satisfied by, e.g., Lipschitz MDPs (Asadi et al., 2018),[2] although this assumption holds much more generally.

## 4 Main Results

In this section, we present our main results. Theorem 1 establishes the non-asymptotic convergence rate of `POLY-HOOT`. Theorem 2 characterizes the concentration rates of regret of enhanced HOO in a non-stationary bandit setting; this result serves as an important intermediate step in the analysis of `POLY-HOOT`. The proofs for Theorems 1 and 2 are given in Appendices C and B, respectively.

### 4.1 Convergence of `POLY-HOOT`

**Theorem 1.** *Consider an MDP that satisfies Assumptions 1 and 2. For any $D \geq 1$, run $n$ rounds of MCTS simulations with parameters specified as follows:*

$$
\begin{aligned}
\alpha^{(d)} &= \left(1 - \eta^{(d)}\right) \eta^{(d)} \xi^{(d)}, & 0 \leq d \leq D - 1, \\
\xi^{(d-1)} &= \left(\alpha^{(d)} - 3\right)/2, & 1 \leq d \leq D - 1, \\
\eta^{(d-1)} &= \frac{\frac{\alpha^{(d)}}{\xi^{(d)}(1-\eta^{(d)})} + d' + \frac{1}{1-\eta^{(d)}}}{1 + d' + \frac{1}{1-\eta^{(d)}}}, & 1 \leq d \leq D - 1,
\end{aligned}
\tag{3}
$$

*where $d' > 0$ is a constant to be specified in Definition 3 (Appendix B). Suppose that $\xi^{(D-1)} > 0$ and $\frac{1}{2} \leq \eta^{(D-1)} < 1$ are chosen large enough such that $\alpha^{(0)} > 3$, and $\bar{H}$ satisfies $\rho^{\bar{H}} < n^{\eta^{(0)}-1}$. Then for each query state $s \in S$, the following result holds for the output $\hat{V}_n(s)$ of Algorithm 1:*

$$\left| \mathbb{E}\left[ \hat{V}_n(s) \right] - V^*(s) \right| \leq O\left( \frac{1}{n^\zeta} \right) + \gamma^D \varepsilon_0,$$

*where $\zeta \in (0, \frac{1}{2})$ satisfies $\zeta \leq 1 - \eta^{(d)}, \forall\, 0 \leq d \leq D - 1$, and $\varepsilon_0 = \left\| \hat{V} - V^* \right\|_\infty$ is the error in the value function oracle at the leaf nodes.*

*Proof Sketch.* MCTS can be viewed as a hierarchy of multi-armed bandits (in our case, continuous-armed bandits), one per each node in the tree. In particular, the rewards of the bandit associated with each intermediate node are the rewards generated by the bandit algorithms for nodes downstream. Since the HOO policy is changing to balance exploitation-exploration, the resulting rewards are non-stationary. With this observation, the proof for Theorem 1 can be broken down to the following three steps:

**1. Non-stationary bandits.** The first step concerns the analysis of a non-stationary bandit, which models the MAB at each node on the MCTS search tree. In particular, we show that if the rewards of a continuous-armed bandit problem satisfy certain convergence and concentration properties, then the regret induced by the enhanced HOO algorithm satisfies similar convergence and concentration guarantees. The result is formally established in Theorem 2.

**2. Induction step.** Since the rewards collected at one level of bandits constitute the bandit rewards of the level above it, we can apply the results of Step 1 recursively, from level $D - 1$ upwards to the root node. We inductively show that the bandit rewards at each level $d$ of MCTS satisfy the properties required by Theorem 2, and hence we can propagate the convergence and concentration properties to the bandit at level $d - 1$, using the results of Theorem 2. The convergence result for the root node is established by induction.

**3. Error from the oracle.** Finally, we consider the error induced by the leaf node estimator, i.e., the value function oracle $\hat{V}$. Given a value function oracle $\hat{V}$ for the leaf nodes, a depth-$D$ MCTS can be effectively viewed as $D$ steps of value iteration starting from $\hat{V}$ (Shah et al., 2019). Therefore, the error in the value function oracle $\hat{V}$ shrinks at a geometric rate of $\gamma$ due to the contraction mapping. $\qquad\square$

Theorem 1 implies that the value function estimate obtained by Algorithm 1 converges to the $\gamma^D \varepsilon_0$-neighborhood of the optimal value function at a rate of $O(n^{-\zeta})$, where $\zeta \in (0, \frac{1}{2})$ depends on the parameters $\alpha^{(D-1)}, \xi^{(D-1)}$, and $\eta^{(D-1)}$ we choose. Therefore, by setting the depth $D$ of MCTS appropriately, Algorithm 1 can output an estimate that is within an arbitrarily small neighborhood around the optimal values.

*Remark* 3. We remark on several technical challenges in the proof of Theorem 1. The first challenge is to transform a hierarchy of inter-dependent bandits into a recursive sequence of non-stationary bandit problems with unified form, which is highly non-trivial even in the finite case (Shah et al., 2019). As far as we know, a general solution to non-stationary bandit problems with continuous domains is not available in the literature. Our enhanced HOO algorithm might be of independent research interest. Another challenge is to ensure sufficient exploitation in face of infinitely many candidate arms and strong non-stationarity of rewards. Existing solutions include uniformly sampling actions through progressive widening (Auger et al., 2013) and playing each action for a fixed amount of times (Kim et al., 2020). Instead, our solution balances the trade-off between exploration and exploitation by using a limited depth HOO bandit, which makes our theoretical analysis highly non-trivial.

## 4.2 Enhanced HOO in the Non-Stationary Setting

The key step in the proof of Theorem 1 is to establish the following result for the enhanced HOO bandit algorithm. Consider a continuous-armed bandit on the domain $X \subseteq [0, 1]^m$, with non-stationary rewards bounded in $[-R, R]$ satisfying the following properties:

A. Fixed-arm convergence: The mean-payoff function $f_n : X \to \mathbb{R}$ converges to a function $f : X \to \mathbb{R}$ in $L^\infty$ at a polynomial rate:

$$\|f_n - f\|_\infty \leq \frac{C}{n^\varsigma}, \ \forall n \geq 1, \tag{4}$$

for some constant $C > 0$ and $0 < \varsigma < \frac{1}{2}$.

B. Fixed-arm concentration: There exist constants $\beta > 1, \xi > 0$, and $1/2 \leq \eta < 1$, such that for every $z \geq 1$ and every integer $n \geq 1$:

$$\mathbb{P}\left(\sum_{t=1}^n X_t - nf(x) \geq n^\eta z\right) \leq \frac{\beta}{z^\xi} \quad \text{and} \quad \mathbb{P}\left(\sum_{t=1}^n X_t - nf(x) \leq -n^\eta z\right) \leq \frac{\beta}{z^\xi}, \ \forall x \in X, \tag{5}$$

where $X_t$ denotes the random reward obtained by pulling arm $x \in X$ for the $t$-th time.

**Theorem 2.** *Consider a non-stationary continuous-armed bandit problem satisfying properties* (4) *and* (5)*. Suppose we apply the enhanced HOO agent defined in Algorithms 2 and 3 with parameters satisfying $\xi\eta(1-\eta) \leq \alpha < \xi(1-\eta)$, $\alpha > 3$, and $\rho^{\bar{H}} < n^{\eta-1}$. Let the random variable $Y_t$ denote the reward obtained at time $t$. Then the following holds:*

A. *Optimal-arm convergence: There exists some constant $C_0 > 0$, such that*

$$\left|\frac{1}{n}\mathbb{E}\left[\sum_{t=1}^n Y_t\right] - f^*\right| \leq \frac{C_0}{n^\varsigma}, \tag{6}$$

*where $0 < \varsigma \leq \frac{1 - \frac{\alpha}{\xi(1-\eta)}}{1+d'+\frac{1}{1-\eta}}$.*

B. *Optimal-arm concentration: There exist constants $\beta' > 1, \xi' > 0$, and $1/2 \leq \eta' < 1$, such that for every $z \geq 1$ and every integer $n \geq 1$:*

$$\mathbb{P}\left(\sum_{t=1}^n Y_t - nf^* \geq n^{\eta'} z\right) \leq \frac{\beta'}{z^{\xi'}} \quad \text{and} \quad \mathbb{P}\left(\sum_{t=1}^n Y_t - nf^* \leq -n^{\eta'} z\right) \leq \frac{\beta'}{z^{\xi'}}, \tag{7}$$

*where $\eta' = \frac{\frac{\alpha}{\xi(1-\eta)}+d'+\frac{1}{1-\eta}}{1+d'+\frac{1}{1-\eta}}$, $\xi' = (\alpha - 3)/2$, and $\beta' > 1$ depends on $\alpha, \beta, \eta, \xi$ and $\bar{H}$.*

Theorem 2 states the properties of the regret induced by the enhanced HOO algorithm (Algorithms 2 and 3) for a non-stationary continuous-armed bandit problem, which may be of independent interest. If the rewards of the non-stationary bandit satisfy certain convergence rate and concentration conditions, then the regret of our algorithm also enjoys the same convergence rate and similar concentration guarantees. We can verify that our configuration of the parameters $\alpha^{(d)}, \xi^{(d)}, \eta^{(d)}, 0 \leq d \leq D-1$ in Theorem 1 satisfy the requirements of Theorem 2. Therefore, using this theorem we can propagate the convergence result on one level of MCTS to the level above it. By applying Theorem 2 recursively, we can establish the convergence result of the value function estimate for the root node of MCTS.

In addition to the technical difficulty of analyzing the regret of HOO (Bubeck et al., 2011), we have to address the challenges raised by the non-stationary rewards and bounded depth of HOO tree. The results are formally established as a sequence of lemmas in Appendix D.

## 5    Simulations

In this section, we empirically evaluate the performance of `POLY-HOOT` on several classic control tasks. We have chosen three benchmark tasks from OpenAI Gym (OpenAI, 2016), and extended them to the continuous-action settings as necessary. These tasks include CartPole, Inverted Pendulum Swing-up, and LunarLander. CartPole is relatively easy, so we have also modified it to a more challenging one, CartPole-IG, with an increased gravity value. This new setting requires smoother actions, and bang-bang control strategies easily cause the pole to fall due to the increased inertia.

We compare the empirical performance of `POLY-HOOT` with three other continuous MCTS algorithms, including UCT (Kocsis and Szepesvári, 2006) with manually discretized actions, Polynomial Upper

| | CartPole | CartPole-IG | Pendulum | LunarLander |
|---|---|---|---|---|
| discretized-UCT | $77.85 \pm 0.0$ | $69.39 \pm 6.63$ | $-109.68 \pm 0.29$ | $-57.95 \pm 77.36$ |
| PUCT | $77.85 \pm 0.0$ | $71.48 \pm 8.27$ | $-109.64 \pm 0.25$ | $-43.05 \pm 80.25$ |
| HOOT | $77.85 \pm 0.0$ | $77.85 \pm 0.0$ | $-109.50 \pm 0.35$ | $-23.37 \pm 76.46$ |
| `POLY-HOOT` | $77.85 \pm 0.0$ | $77.85 \pm 0.0$ | $-109.43 \pm 0.25$ | $-3.02 \pm 44.41$ |

Table 1: Empirical performances on classic control tasks

| Algorithm | discretized-UCT | PUCT | HOOT | $\bar{H} = 2$ | $\bar{H} = 4$ | $\bar{H} = 6$ | $\bar{H} = 8$ | $\bar{H} = 10$ |
|---|---|---|---|---|---|---|---|---|
| Reward | 69.03 | 70.79 | 77.85 | 42.45 | 48.54 | 63.27 | 77.85 | 77.85 |
| Time per decision (s) | 0.950 | 0.305 | 1.173 | 0.054 | 0.149 | 0.610 | 1.030 | 1.057 |

Table 2: Time per decision on CartPole-IG

Confidence Trees (PUCT) with progressive widening (Auger et al., 2013), and the original implementation of HOOT (Mansley et al., 2011) with a logarithmic bonus term. Their average rewards and standard deviations on the above tasks are shown in Table 1. The results are averaged over 40 runs. The detailed experiment settings as well as additional experiment results can be found in Appendix E.

As we can see from Table 1, all four algorithms achieve optimal rewards on the easier CartPole task. However, for the CartPole-IG task with increased gravity, discretized-UCT and PUCT do not achieve the optimal performance, because their actions, either sampled from a uniform grid or sampled completely randomly, are not smooth enough to handle the larger momentum. In the Pendulum task, the four algorithms have similar performance, although HOOT and `POLY-HOOT` perform slightly better. Finally, on LunarLander, HOOT and `POLY-HOOT` achieve much better performances. This task has a high-dimensional action space, making it difficult for discretized-UCT and PUCT to sample actions at fine granularity. Also note that `POLY-HOOT` significantly outperforms HOOT. We believe the reason is that this task, as detailed in Appendix E, features a deeper search depth and sparse but large positive rewards. This causes a more severe non-stationarity issue of rewards within the search tree, which is better handled by `POLY-HOOT` with a polynomial bonus term than by HOOT, as our theory suggests. This demonstrates the superiority of `POLY-HOOT` in dealing with complicated continuous-space tasks with higher dimensions and deeper planning depth. We would also like to remark that the high standard deviations in this task are mostly due to the reward structure of the task itself—the agent either gets a large negative reward (when the lander crashes) or a large positive reward (when it lands on the landing pad) in the end.

We also empirically evaluate the time complexity of the algorithms. Table 2 shows the time needed by each algorithm to make a single decision on CartPole-IG. For `POLY-HOOT`, we further test its computation time with different values of $\bar{H}$ (the maximum depth of the HOO tree), which is an important hyper-parameter to balance the trade-off between optimality and time complexity. All tests are averaged over 10 (new) runs on a laptop with an Intel Core i5-9300H CPU. We can see that `POLY-HOOT` requires slightly more computation than discretized-UCT and PUCT as the cost of higher rewards, but it is still more time-efficient than HOOT because of the additional depth limitation.

## 6 Conclusions

In this paper, we have considered Monte-Carlo planning in an environment with continuous state-action spaces. We have introduced `POLY-HOOT`, an algorithm that augments MCTS with a continuous armed bandit strategy HOO. We have enhanced HOO with an appropriate polynomial bonus term in the upper confidence bounds, and investigated the regret of the enhanced HOO algorithm in non-stationary bandit problems. Based on this result, we have established non-asymptotic convergence guarantees for `POLY-HOOT`. Experimental results have further corroborated our theoretical findings. Our theoretical results have advocated the use of non-stationary bandits with polynomial bonus terms in MCTS, which might guide the design of new planning algorithms in continuous spaces, with potential applications in robotics and control, that enjoy better empirical performance as well.

## Broader Impact

We believe that researchers of planning, reinforcement learning, and multi-armed bandits, especially those who are interested in the theoretical foundations, would benefit from this work. In particular, prior to this work, though intuitive, easy-to-implement, and empirically widely-used, a theoretical

analysis of Monte-Carlo tree search (MCTS) in continuous domains had not been established through the lens of non-stationary bandits. In this work, inspired by the recent advances in finite-space Monte-Carlo tree search, we have provided such a result, and thus theoretically justified the efficiency of MCTS in continuous domains.

Although Monte-Carlo tree search has demonstrated great performance in a wide range of applications, theoretical explanation of its empirical successes is relatively lacking. Our theoretical results have advocated the use of non-stationary bandit algorithms, which might guide the design of new planning algorithms that enjoy better empirical performance in practice. Our results might also be helpful for researchers interested in robotics and control applications, as our algorithm can be readily applied to such planning problems with continuous domains.

As a theory-oriented work, we do not believe that our research will cause any ethical issue, or put anyone at any disadvantage.

## Acknowledgments and Disclosure of Funding

We thank Bin Hu for helpful comments on an earlier version of the paper. Research of the three authors from Illinois was supported in part by Office of Naval Research (ONR) MURI Grant N00014-16-1-2710, and in part by the US Army Research Laboratory (ARL) Cooperative Agreement W911NF-17-2-0196. Q. Xie is partially supported by NSF grant 1955997.

## Footnotes

[1]We use $h$ and $H$ to index the depth in the HOO tree, and use $d$ and $D$ to index the depth in the MCTS tree.

[2]This is the class of MDPs whose reward functions and (possibly deterministic) state transitions satisfy certain smoothness criteria with respect to, say, the Wasserstein metric. As observed in Asadi et al. (2018), the Wasserstein metric is often more appropriate than the Kullback-Leibler divergence metric in Lipschitz MDPs.

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
