[Supplementary Material]

# Supplementary Materials for "`POLY-HOOT`: Monte-Carlo Planning in Continuous Space MDPs with Non-Asymptotic Analysis"

## A  Algorithm Details

In the following, we provide the details of the functions $HOO\_query$ and $HOO\_update$ that are utilized in Algorithm 1.

---

**Algorithm 2:** HOO_query

---

1 **Input:** depth in MCTS $d$, state $s$, and round $t$.
2 **Output:** action to take $a$.
3 **Parameters:** maximum depth $\bar{H}$ allowed in HOO.
4 **if** *state $s$ has never been visited at MCTS depth $d$* **then**
5      Initialize HOO agent at state $s$ and depth $d$: $\mathcal{T} \leftarrow \{(0,1)\}$ and $B_{1,2}, B_{2,2} \leftarrow \infty$;
6 **else**
7      $\mathcal{T} \leftarrow$ the HOO agent constructed at state $s$ and depth $d$ previously;
8 $(h, i) \leftarrow (0, 1)$;
9 Initialize HOO path in the current round: $P_t \leftarrow \{(h, i)\}$;
10 **while** $(h, i) \in \mathcal{T}$ **do**
11      **if** $B_{h+1, 2i-1} > B_{h+1, 2i}$ **then**
12          $(h, i) \leftarrow (h+1, 2i-1)$;
13      **else**
14          $(h, i) \leftarrow (h+1, 2i)$;
15      $P_t \leftarrow P_t \cup \{(h, i)\}$
16 $(H, I) \leftarrow (h, i)$;
17 **if** $H \le \bar{H}$ **then**
18      Choose arbitrary arm $X$ in $\mathcal{P}_{H,I}$;
19      $A_{H,I} = X$;
     // Associate the chosen action $X$ with the node $(H, I)$.
20      $\mathcal{T} \leftarrow \mathcal{T} \cup \{(H, I)\}$;
21      $B_{H+1, 2I-1}, B_{H+1, 2I} \leftarrow \infty$;
22      **return** $X$;
23 **else**
     // We reached the maximum depth and should not explore new actions.
24      $(H, I) \leftarrow (H-1, \lceil I/2 \rceil)$;
25      **return** $A_{H,I}$.

---

**Algorithm 3:** HOO_update

---

1 **Input:** depth in MCTS $d$, state $s$, and bandit reward $Y$ at round $t$.
2 **Parameters:** $\alpha^{(d)}, \xi^{(d)}, \eta^{(d)}, \nu_1$ and $\rho$.
3 $\alpha, \xi, \eta \leftarrow \alpha^{(d)}, \xi^{(d)}, \eta^{(d)}$;
4 **foreach** *$(h, i)$ in $P_t$* **do**
5      $T_{h,i} \leftarrow T_{h,i} + 1$;
6      $\widehat{\mu}_{h,i} \leftarrow (1 - 1/T_{h,i})\,\widehat{\mu}_{h,i} + Y/T_{h,i}$;
7 **foreach** *$(h, i)$ in $\mathcal{T}$* **do**
8      $U_{h,i} \leftarrow \widehat{\mu}_{h,i} + t^{\alpha/\xi} T_{h,i}^{\eta-1} + \nu_1 \rho^h$;
9 $\mathcal{T}' \leftarrow \mathcal{T}$;
10 **while** $\mathcal{T}' \neq \{(0,1)\}$ **do**
11      $(h, i) \leftarrow$ an arbitrary leaf node of $\mathcal{T}'$;
12      $B_{h,i} \leftarrow \min\{U_{h,i}, \max\{B_{h+1,2i-1}, B_{h+1,2i}\}\}$;
13      $\mathcal{T}' \leftarrow \mathcal{T}' \backslash \{(h, i)\}$;

---

# B  Proof of Theorem 2

Let $R_n = \sum_{t=1}^{n}(f^* - Y_t)$ denote the regret of Algorithms 2 and 3 with the depth limitation $\bar{H}$. We define the following notations that are similar to Bubeck et al. (2011). First, let $I_h$ denote the set of nodes at depth $h$ that are $2\nu_1\rho^h$-optimal, i.e., the set of nodes $(h,i)$ that satisfy $f^*_{h,i} \geq f^* - 2\nu_1\rho^h$, where $f^*_{h,i} \triangleq \sup_{x\in\mathcal{P}_{h,i}} f(x)$. For $h \geq 1$, let $J_h$ denote the set of nodes at depth $h$ that are not in $I_h$ but whose parents are in $I_{h-1}$ (i.e., they are not $2\nu_1\rho^h$-optimal themselves but their parents are $2\nu_1\rho^{h-1}$-optimal). Finally, define $\mathcal{X}_\varepsilon \triangleq \{x \in X : f(x) \geq f^* - \varepsilon\}$ to be the set of arms that are $\varepsilon$-close to optimal.

Let $(H_t, I_t)$ denote the node that is selected by the bandit algorithm at time $t$. Note that with the depth limitation $\bar{H}$ it is possible that the nodes on depth $\bar{H}$ might be played more than once at different rounds. The nodes above depth $\bar{H}$ (i.e., $H_t < \bar{H}$), on the other hand, are played only once and the random variables $(H_t, I_t)$ are not the same for different values of $t$. Let $\mathcal{L} = \{(H_t, I_t) : H_t = \bar{H}\}$ denote the set of nodes on depth $\bar{H}$ that have been played. Let $H \geq 1$ be a constant integer whose value will be specified later, and without loss of generality we assume $\bar{H} > H$. We partition the nodes in the HOO tree $\mathcal{T}$ above depth $\bar{H}$ into three parts $\mathcal{T}\backslash\mathcal{L} = \mathcal{T}_1 \cup \mathcal{T}_2 \cup \mathcal{T}_3$. Let $\mathcal{T}_1$ be the set of nodes above depth $\bar{H}$ that are descendants of nodes in $I_H$. By convention, a node itself is also considered as a descendant of its own, so we also have $I_H \subseteq \mathcal{T}_1$. Let $\mathcal{T}_2 = \cup_{0\leq h<H} I_h$. Finally, let $\mathcal{T}_3$ be the set of nodes above depth $\bar{H}$ that are descendants of nodes in $\cup_{0\leq h\leq H} J_h$. We can verify that $\mathcal{T}_1 \cup \mathcal{T}_2 \cup \mathcal{T}_3 \cup \mathcal{L}$ covers all the nodes in $\mathcal{T}$.

Similarly, we also decompose the regret according to the selected node $(H_t, I_t)$ into four parts: $R_n = R_{n,1} + R_{n,2} + R_{n,3} + R_\mathcal{L}$, where $R_{n,i} = \sum_{t=1}^{n}(f^* - Y_t)\mathbb{I}_{\{(H_t,I_t)\in\mathcal{T}_i\}}$ and $R_\mathcal{L} = \sum_{t=1}^{n}(f^* - Y_t)\mathbb{I}_{\{(H_t,I_t)\in\mathcal{L}\}}$. In the following, we analyze each of the four parts individually. We start with the concentration property and then the convergence results.

To proceed further, we first need to state several definitions that are useful throughout. These definitions come from Bubeck et al. (2011), with similar ideas introduced earlier in Auer et al. (2007). We reproduce these definitions here for completeness.

**Definition 1.** *(Packing number) The $\varepsilon$-packing number $\mathcal{N}(\mathcal{X}, \ell, \varepsilon)$ of $\mathcal{X}$ w.r.t the dissimilarity $\ell$ is the largest integer $k$ such that there exists $k$ disjoint $\ell$-open balls with radius $\varepsilon$ contained in $\mathcal{X}$.*

**Definition 2.** *(Near-optimality dimension) For $c > 0$, the near-optimality dimension of $f$ w.r.t $\ell$ is*

$$\max\left\{0, \limsup_{\varepsilon\to 0} \frac{\ln \mathcal{N}(\mathcal{X}_{c\varepsilon}, \ell, \varepsilon)}{\ln(\varepsilon^{-1})}\right\}.$$

**Definition 3.** *Let $d$ be the $4\nu_1/\nu_2-$near-optimality dimension of $f$ w.r.t $\ell$. We use $d'$ to denote any value such that $d' > d$.*

**Definition 4.** *Given the limit of the mean-payoff function $f$ of a HOO agent, we assume without loss of generality that $(0,1), (1,i_1^*), (2,i_2^*), \ldots, (\bar{H}, i_{\bar{H}}^*)$ is an optimal path, i.e., $\Delta_{h,i_h^*} = 0, \forall h \geq 1$. We define the nodes $(h, i_h^*)$ on the optimal path as optimal nodes, and the other nodes as suboptimal nodes.*

Our proof will also rely on several lemmas that we state and prove in Appendix D.

## B.1  Regret from $\mathcal{T}_1$

Any node in $I_H$ is by definition $2\nu_1\rho^H$-optimal. By Lemma 2, the domain of $I_H$ lies in $\mathcal{X}_{4\nu_1\rho^H}$. Since the descendants of $I_H$ cover a domain that is a subset of the domain of $I_H$, we know the descendants of $I_H$ also lie in the domain of $\mathcal{X}_{4\nu_1\rho^H}$, and hence $\sum_{t=1}^{n}(f^* - f(X_t))\mathbb{I}_{\{(H_t,I_t)\in\mathcal{T}_1\}} \leq 4\nu_1\rho^H n$.

Let $n_1 = |\mathcal{T}_1|$ we then have for every $z \geq 1$,

$$
\begin{aligned}
&\mathbb{P}\left(R_{n,1} \geq z n^\eta + 4\nu_1 \rho^H n\right) \\
=&\mathbb{P}\left(\sum_{t=1}^n (f^* - Y_t)\,\mathbb{I}_{\{(H_t, I_t) \in \mathcal{T}_1\}} \geq z n^\eta + 4\nu_1 \rho^H n\right) \\
=&\mathbb{P}\left(\sum_{t=1}^n (f^* - f(X_t))\,\mathbb{I}_{\{(H_t, I_t) \in \mathcal{T}_1\}} + \sum_{t=1}^n (f(X_t) - Y_t)\,\mathbb{I}_{\{(H_t, I_t) \in \mathcal{T}_1\}} \geq z n^\eta + 4\nu_1 \rho^H n\right) \\
\leq&\sum_{t=1}^{n_1} \mathbb{P}\left(f(\tilde{X}_t) - \tilde{Y}_t \geq \frac{z}{n_1} n^\eta\right) \\
\leq&\frac{n_1^{\xi+1}\beta}{z^\xi} \leq \frac{c_1^{\xi+1}\beta}{z^{\alpha-3}},
\end{aligned}
$$

where $\tilde{X}_t$ denotes the $t$-th arm pulled in $\mathcal{T}_1$, and $\tilde{Y}_t$ denotes its corresponding reward. Note that in the first inequality we used the fact that $\sum_{t=1}^n (f^* - f(X_t))\,\mathbb{I}_{\{(H_t, I_t) \in \mathcal{T}_1\}} \leq 4\nu_1 \rho^H n$. In the second inequality we used the union bound. In the third inequality we applied the concentration property of the bandit problem (5) with $n = 1$. Notice that we can only use the concentration property when the requirement $\frac{z}{n_1} \geq 1$ is satisfied, but when $\frac{z}{n_1} < 1$, the inequality also trivially holds because $\frac{n_1^{\xi+1}\beta}{z^\xi} > 1$. The last step holds because $\alpha - 3 < \alpha < \xi(1-\eta) < \xi$, and $c_1 \geq 1$ is a constant that upper bounds $n_1$ (since $\mathcal{T}$ is a binary tree with limited depth, one trivial upper bound would be the number of nodes in $\mathcal{T}$, which does not depend on $n$ and $z$). Also notice that the inequality above trivially holds when $0 < z < 1$, because $\beta > 1, \alpha - 3 > 0$ and hence $\frac{\beta}{z^{\alpha-3}} > 1$ is an upper bound for any probability value.

Let $\lambda = \frac{\frac{\alpha}{\xi(1-\eta)} - 1}{1 + d' + \frac{1}{1-\eta}}$, and we know $\lambda < 0$ because $\alpha < \xi(1-\eta)$. We then choose the value for $H$ such that $\rho^H = n^\lambda$; then, $4\nu_1 \rho^H n$ is of the order of $n^{\lambda+1}$. We further have $n^{\lambda+1} > n^\eta$ since $\alpha \geq \xi\eta(1-\eta)$. Let $c_2 \geq 1$ be a constant such that $c_2 n^{\lambda+1} \geq c_2^{1/2} n^\eta + 4\nu_1 n^{\lambda+1}, \forall n \geq 1$. Such a constant always exists because $c_2^{1/2} < c_2$ and $n^\eta < n^{\lambda+1}$. Then it is easy to see that $z n^{\lambda+1} \geq z^{1/2} n^\eta + 4\nu_1 n^{\lambda+1}, \forall n \geq 1$ also holds for any $z \geq c_2$. Therefore, we have the following property:

$$
\mathbb{P}\left(R_{n,1} \geq z n^{\lambda+1}\right) \leq \frac{c_1^{\xi+1} c_2^{\alpha-3}\beta}{z^{(\alpha-3)/2}}, \quad \forall z \geq 1. \tag{8}
$$

To see this, first suppose that $z \geq c_2$; then, $z n^{\lambda+1} \geq z^{1/2} n^\eta + 4\nu_1 n^{\lambda+1}, \forall n \geq 1$ and since $c_2 \geq 1$, we have $\mathbb{P}\left(R_{n,1} \geq z n^{\lambda+1}\right) \leq \mathbb{P}\left(R_{n,1} \geq \frac{z^{1/2}}{c_2} n^\eta + 4\nu_1 \rho^H n\right) \leq \frac{c_1^{\xi+1} c_2^{\alpha-3}\beta}{z^{(\alpha-3)/2}}$. On the other hand, if $1 \leq z < c_2$, then the inequality (8) trivially holds, because $c_2^{\alpha-3} > z^{\alpha-3} \geq z^{(\alpha-3)/2}$ and $\beta > 1, c_1 \geq 1$, making the RHS greater than 1. The other side of the concentration inequality follows similarly and is omitted here.

## B.2 Regret from $\mathcal{T}_2$

For $h \geq 0$, any node $(h, i) \in \mathcal{T}_2$ by definition belongs to $I_h$ and is hence $2\nu_1 p^h$-optimal. Therefore, $\sum_{t=1}^n (f^* - f(X_t))\,\mathbb{I}_{\{(H_t, I_t) \in \mathcal{T}_2\}} \leq \sum_{h=0}^{H-1} 4\nu_1 \rho^h |I_h| \leq 4c_3\nu_1\nu_2^{-d'}\sum_{h=0}^{H-1} \rho^{h(1-d')}$, where the last step uses the fact that $|I_h| \leq c_3\left(\nu_2\rho^h\right)^{-d'}$ for some constant $c_3$ (Lemma 3 in Appendix D). We then have the following convergence result:

$$
\mathbb{E}\left[R_{n,2}\right] \leq 4c_3\nu_1\nu_2^{-d'}\sum_{h=0}^{H-1} \rho^{h(1-d')}. \tag{9}
$$

Let $n_2 = |\mathcal{T}_2|$; then for every $z \geq 1$, we have

$$\mathbb{P}\left(R_{n,2} \geq zn^\eta + 4c_3\nu_1\nu_2^{-d'}\sum_{h=0}^{H-1}\rho^{h(1-d')}\right)$$

$$=\mathbb{P}\left(\sum_{t=1}^{n}(f^* - Y_t)\,\mathbb{I}_{\{(H_t,I_t)\in\mathcal{T}_2\}} \geq zn^\eta + 4c_3\nu_1\nu_2^{-d'}\sum_{h=0}^{H-1}\rho^{h(1-d')}\right)$$

$$=\mathbb{P}\left(\sum_{t=1}^{n}(f^* - f(X_t))\,\mathbb{I}_{\{(H_t,I_t)\in\mathcal{T}_2\}} + \sum_{t=1}^{n}(f(X_t) - Y_t)\,\mathbb{I}_{\{(H_t,I_t)\in\mathcal{T}_2\}}\right.$$
$$\left. \geq zn^\eta + 4c_3\nu_1\nu_2^{-d'}\sum_{h=0}^{H-1}\rho^{h(1-d')}\right)$$

$$\leq\mathbb{P}\left(\sum_{t=1}^{n}(f(X_t) - Y_t)\,\mathbb{I}_{\{(H_t,I_t)\in\mathcal{T}_2\}} \geq zn^\eta\right)$$

$$\leq\frac{n_2^{\xi+1}\beta}{z^\xi} \leq \frac{c_4^{\xi+1}\beta}{z^{\alpha-3}},$$

where the first inequality uses the fact that $\sum_{t=1}^{n}(f^* - f(X_t))\,\mathbb{I}_{\{(H_t,I_t)\in\mathcal{T}_2\}} \leq 4c_3\nu_1\nu_2^{-d'}\sum_{h=0}^{H-1}\rho^{h(1-d')}$, and $c_4$ is a constant not depending on $n$ and $z$ that upper bounds $n_2$, similar to the proof in $\mathcal{T}_1$. Again, this inequality also trivially holds for $0 < z < 1$.

Since there exists a constant $c_5$ that

$$\sum_{h=0}^{H-1}\rho^{h(1-d')} \leq c_5\rho^{H(1-d')} \leq c_5\rho^{-H(d'+\frac{1}{1-\eta})} \leq c_5\rho^{-H(d'+\frac{1}{1-\eta})}n^{\frac{\alpha}{\xi(1-\eta)}} \leq c_5 n^{\lambda+1},$$

we know $4c_3\nu_1\nu_2^{-d'}\sum_{h=0}^{H-1}\rho^{h(1-d')}$ is upper bounded by the order of $n^{\lambda+1}$. Again, since $n^{\lambda+1} > n^\eta$, there always exists a constant $c_6 \geq 1$ such that for any $z \geq c_6$, $zn^{\lambda+1} \geq z^{1/2}n^\eta + 4c_3\nu_1\nu_2^{-d'}\sum_{h=0}^{H-1}\rho^{h(1-d')}, \forall n \geq 1$. Therefore, we have

$$\mathbb{P}\left(R_{n,2} \geq zn^{\lambda+1}\right) \leq \frac{c_4^{\xi+1}c_6^{\alpha-3}\beta}{z^{(\alpha-3)/2}}, \ \forall z \geq 1. \tag{10}$$

To see this, again, first suppose that $z \geq c_6$, then $zn^{\lambda+1} \geq z^{1/2}n^\eta + 4c_3\nu_1\nu_2^{-d'}\sum_{h=0}^{H-1}\rho^{h(1-d')}$, and hence $\mathbb{P}\left(R_{n,2} \geq zn^{\lambda+1}\right) \leq \mathbb{P}\left(R_{n,2} \geq \frac{z^{1/2}}{c_6}n^\eta + 4c_3\nu_1\nu_2^{-d'}\sum_{h=0}^{H-1}\rho^{h(1-d')}\right) \leq \frac{c_4^{\xi+1}c_6^{\alpha-3}\beta}{z^{(\alpha-3)/2}}$. If on the other hand $1 \leq z < c_6$, then inequality (10) trivially holds because the RHS is greater than 1.

### B.3  Regret from $\mathcal{T}_3$

For any node $(h,i) \in \mathcal{T}_3$, since the parent of any $(h,i) \in J_h$ is in $I_{h-1}$, we know by Lemma 2 that the domain of $(h,i)$ is in $\mathcal{X}_{4\nu_1\rho^{h-1}}$. Further, for any $u \geq A_{h,i}(n) = \left\lceil\left(\frac{2n^{\alpha/\xi}}{\Delta_{h,i}-\nu_1\rho^h}\right)^{\frac{1}{1-\eta}}\right\rceil$ and $z \geq 1$, we know from inequality (21) that $\mathbb{P}\left(T_{h,i}(n) > zu\right) \leq \frac{(zu-1)^{3-\alpha}}{n} + \frac{(zu-1)^{3-\alpha}}{\alpha-3} \leq z^{3-\alpha}(u-1)^{3-\alpha}\left(\frac{1}{n} + \frac{1}{\alpha-3}\right)$. Since $\Delta_{h,i} > 2\nu_1\rho^h$, we know $A_{h,i}(n) \leq \left\lceil\left(\frac{2n^{\alpha/\xi}}{\nu_1\rho^h}\right)^{\frac{1}{1-\eta}}\right\rceil$. Then for

any $u > \left(\frac{2n^{\alpha/\xi}}{\nu_1 \rho^h}\right)^{\frac{1}{1-\eta}}$,

$$\mathbb{P}\left(\sum_{t=1}^{n}\left(f^* - f\left(X_t\right)\right)\mathbb{I}_{\{(H_t, I_t)\in\mathcal{T}_3\}} \geq \sum_{h=1}^{H}4\nu_1\rho^{h-1}\sum_{(h,i)\in\mathcal{T}_3}zu\right)$$

$$\leq\mathbb{P}\left(\sum_{h=1}^{H}4\nu_1\rho^{h-1}\sum_{(h,i)\in\mathcal{T}_3}T_{h,i}(n) \geq \sum_{h=1}^{H}4\nu_1\rho^{h-1}\sum_{(h,i)\in\mathcal{T}_3}zu\right)$$

$$\leq\sum_{h=1}^{H}\mathbb{P}\left(\sum_{(h,i)\in\mathcal{T}_3}T_{h,i}(n) \geq \sum_{(h,i)\in\mathcal{T}_3}zu\right)$$

$$\leq\sum_{h=1}^{H}|J_h|z^{3-\alpha}(u-1)^{3-\alpha}\left(\frac{1}{n} + \frac{1}{\alpha-3}\right)$$

$$\leq 2C\nu_2^{-d'}\sum_{h=1}^{H}\rho^{-(h-1)d'}z^{3-\alpha}(u-1)^{3-\alpha}\left(\frac{1}{n} + \frac{1}{\alpha-3}\right),$$

where in the last step we used the fact that $|J_h| \leq 2|I_{h-1}| \leq 2c_2\left(\nu_2\rho^{h-1}\right)^{-d'}$, because the parent of any node in $J_h$ is in $I_{h-1}$. Since $\alpha > 3$, we know $2c_2\nu_2^{-d'}\sum_{h=1}^{H}\rho^{-(h-1)d'}(u-1)^{3-\alpha}\left(\frac{1}{n} + \frac{1}{\alpha-3}\right)$ decreases polynomially in $n$, and hence there exists some constant $c_7 > 1$, such that $2c_2\nu_2^{-d'}\sum_{h=1}^{H}\rho^{-(h-1)d'}(u-1)^{3-\alpha}\left(\frac{1}{n} + \frac{1}{\alpha-3}\right) \leq c_7$, $\forall n \geq 1$. Therefore, for any $z \geq 1$,

$$\mathbb{P}\left(\sum_{t=1}^{n}\left(f^* - f\left(X_t\right)\right)\mathbb{I}_{\{(H_t, I_t)\in\mathcal{T}_3\}} \geq \sum_{h=1}^{H}4\nu_1\rho^{h-1}\sum_{(h,i)\in\mathcal{T}_3}zu\right) \leq c_7 z^{3-\alpha}.$$

Let $n_3 = |\mathcal{T}_3|$, and let $\mathbb{I}_{\{\cdot\}}$ denote $\mathbb{I}_{\{(H_t, I_t)\in\mathcal{T}_3\}}$ for short; then for every $z \geq 1$, we have

$$\mathbb{P}\left(R_{n,3} \geq zn^\eta + \sum_{h=1}^{H}4\nu_1\rho^{h-1}\sum_{(h,i)\in\mathcal{T}_3}zu\right)$$

$$=\mathbb{P}\left(\sum_{t=1}^{n}\left(f^* - Y_t\right)\mathbb{I}_{\{(H_t, I_t)\in\mathcal{T}_3\}} \geq zn^\eta + \sum_{h=1}^{H}4\nu_1\rho^{h-1}\sum_{(h,i)\in\mathcal{T}_3}zu\right)$$

$$=\mathbb{P}\left(\sum_{t=1}^{n}\left(f^* - f\left(X_t\right)\right)\mathbb{I}_{\{\cdot\}} + \sum_{t=1}^{n}\left(f(X_t) - Y_t\right)\mathbb{I}_{\{\cdot\}} \geq zn^\eta + \sum_{h=1}^{H}4\nu_1\rho^{h-1}\sum_{(h,i)\in\mathcal{T}_3}zu\right)$$

$$\leq\mathbb{P}\left(\sum_{t=1}^{n}\left(f(X_t) - Y_t\right)\mathbb{I}_{\{\cdot\}} \geq zn^\eta\right) + \mathbb{P}\left(\sum_{t=1}^{n}\left(f^* - f\left(X_t\right)\right)\mathbb{I}_{\{\cdot\}} \geq \sum_{h=1}^{H}4\nu_1\rho^{h-1}\sum_{(h,i)\in\mathcal{T}_3}zu\right)$$

$$=\frac{n_3^{\xi+1}\beta}{z^\xi} + \mathbb{P}\left(\sum_{t=1}^{n}\left(f^* - f\left(X_t\right)\right)\mathbb{I}_{\{\cdot\}} \geq \sum_{h=1}^{H}4\nu_1\rho^{h-1}\sum_{(h,i)\in\mathcal{T}_3}zu\right)$$

$$\leq\frac{c_8^{\xi+1}\beta}{z^\xi} + c_7 z^{3-\alpha} \leq \frac{c_8^{\xi+1}\beta + c_7}{z^{\alpha-3}},$$

where as before $c_8$ is a constant not depending on $n$ and $z$ that upper bounds $n_3$, and in the last step we used the fact that $\alpha - 3 < \alpha < \xi(1-\eta) < \xi$.

Once again, since $\sum_{h=1}^{H}4\nu_1\rho^{h-1}\sum_{(h,i)\in\mathcal{T}_3}u$ is upper bounded by the order of $n^{\lambda+1}$, there exists a constant $c_9 \geq 1$ such that for any $z \geq c_9$, $zn^{\lambda+1} \geq z^{1/2}n^\eta + \sum_{h=1}^{H}4\nu_1\rho^{h-1}\sum_{(h,i)\in\mathcal{T}_3}z^{1/2}u, \forall n \geq$

1. Therefore, we have

$$\mathbb{P}\left(R_{n,3} \geq zn^{\lambda+1}\right) \leq \frac{c_9^{\alpha-3}(c_8^{\xi+1}\beta + c_7)}{z^{(\alpha-3)/2}}, \ \forall z \geq 1, \tag{11}$$

due to exactly the same logic as in $\mathcal{T}_1$ and $\mathcal{T}_2$, by discussing the two cases $z \geq c_9$ and $1 \leq z < c_9$.

## B.4 Regret from $\mathcal{L}$

Recall that $\mathcal{L}$ is the set of nodes that are played on depth $\bar{H}$. We divide the nodes in $\mathcal{L}$ into two parts $\mathcal{L} = \mathcal{L}_1 \cup \mathcal{L}_3$, in analogy to $\mathcal{T}_1$ and $\mathcal{T}_3$ in $\mathcal{T}\backslash\mathcal{L}$. Let $\mathcal{L}_1$ be the set of nodes on depth $\bar{H}$ that are descendants of nodes in $I_H$, and let $\mathcal{L}_3$ be the set of nodes in $\mathcal{L}$ that are descendants of nodes in $\cup_{0\leq h \leq H} J_h$. By the assumption that $\bar{H} > H$, there is no counterpart of $\mathcal{T}_2 = \cup_{0 \leq h < H} I_h$ in $\mathcal{L}$.

Similarly, we also decompose the regret from $\mathcal{L}$ according to the selected node $(H_t, I_t)$ into two parts: $R_{\mathcal{L}} = \tilde{R}_{n,1} + \tilde{R}_{n,3}$, where $\tilde{R}_{n,i} = \sum_{t=1}^{n} (f^* - Y_t)\,\mathbb{I}_{\{(H_t,I_t)\in\mathcal{L}_i\}}$. Analyzing the regret from $\mathcal{L}_1$ and $\mathcal{L}_3$ is almost the same as $\mathcal{T}_1$ and $\mathcal{T}_3$, with only one difference that each node in $\mathcal{L}$ might be played multiple times. We demonstrate with $\mathcal{L}_1$ in the following and the analysis for $\mathcal{L}_3$ naturally follows.

Again, any node in $I_H$ is by definition $2\nu_1\rho^H$-optimal. By Lemma 2, the domain of $I_H$ lies in $\mathcal{X}_{4\nu_1\rho^H}$, and we know the descendants of $I_H$ also lie in the domain of $\mathcal{X}_{4\nu_1\rho^H}$, satisfying $\sum_{t=1}^{n} (f^* - f(X_t))\,\mathbb{I}_{\{(H_t,I_t)\in\mathcal{L}_1\}} \leq 4\nu_1\rho^H n$. Let $\tilde{n}_1 = |\mathcal{L}_1|$. Let $\tilde{X}_1, \ldots, \tilde{X}_{n_1}$ denote the arms pulled in $\mathcal{L}_1$ (we know from Algorithm 2 that only one arm in a node will be played and associated with that node, and this arm will be played repeatedly thereafter). For $j = 1, \ldots, n_1$, define $K_j$ to be the total number of times arm $\tilde{X}_j$ has been played. Finally, let $\tilde{Y}_j^t$ ($1 \leq t \leq K_j$) denote the corresponding reward when the $t$-th time arm $\tilde{X}_j$ is played. Then for every $z \geq 1$,

$$\mathbb{P}\left(\tilde{R}_{n,1} \geq zn^\eta + 4\nu_1\rho^H n\right)$$

$$=\mathbb{P}\left(\sum_{t=1}^{n} (f^* - Y_t)\,\mathbb{I}_{\{(H_t,I_t)\in\mathcal{L}_1\}} \geq zn^\eta + 4\nu_1\rho^H n\right)$$

$$=\mathbb{P}\left(\sum_{t=1}^{n} (f^* - f(X_t))\,\mathbb{I}_{\{(H_t,I_t)\in\mathcal{L}_1\}} + \sum_{t=1}^{n} (f(X_t) - Y_t)\,\mathbb{I}_{\{(H_t,I_t)\in\mathcal{L}_1\}} \geq zn^\eta + 4\nu_1\rho^H n\right)$$

$$\leq\mathbb{P}\left(\sum_{t=1}^{n} (f(X_t) - Y_t)\,\mathbb{I}_{\{(H_t,I_t)\in\mathcal{L}_1\}} \geq zn^\eta\right)$$

$$\leq\sum_{j=1}^{n_1}\mathbb{P}\left(\sum_{t=1}^{K_j} \left(f(\tilde{X}_j) - \tilde{Y}_j^t\right) \geq \frac{z}{\tilde{c}_1}K_j^\eta\right)$$

$$\leq\frac{\tilde{c}_1^{\xi+1}\beta}{z^\xi} \leq \frac{\tilde{c}_1^{\xi+1}\beta}{z^{\alpha-3}},$$

where $\tilde{c}_1 \geq n_1$ is a constant that is independent of $n$ and $z$, and hence $\sum_{j=1}^{n_1}\frac{z}{\tilde{c}_1}K_j^\eta \leq \frac{z}{n_1}\sum_{j=1}^{n_1} n^\eta \leq zn^\eta$. Note that in the first inequality we used the fact that $\sum_{t=1}^{n} (f^* - f(X_t))\,\mathbb{I}_{\{(H_t,I_t)\in\mathcal{T}_1\}} \leq 4\nu_1\rho^H n$. In the second inequality, we used the union bound. In the third inequality we applied the concentration property of the bandit problem (5) with $n = K_j$. Notice that we can only use the concentration property when the requirement $\frac{z}{\tilde{c}_1} \geq 1$ is satisfied, but when $\frac{z}{\tilde{c}_1} < 1$, the inequality also trivially holds because $\frac{\tilde{c}_1^{\xi+1}\beta}{z^\xi} > 1$. The last step holds because $\alpha - 3 < \alpha < \xi(1-\eta) < \xi$. Also notice that the inequality above trivially holds when $0 < z < 1$, because $\beta > 1, \alpha - 3 > 0$ and hence $\frac{\beta}{z^{\alpha-3}} > 1$ is an upper bound for any probability.

Similar to the analysis of $\mathcal{T}_1$, let $\tilde{c}_2 \geq 1$ be a constant such that $\tilde{c}_2 n^{\lambda+1} \geq \tilde{c}_2^{1/2}n^\eta + 4\nu_1 n^{\lambda+1}, \forall n \geq 1$. Such a constant always exists because $\tilde{c}_2^{1/2} < \tilde{c}_2$ and $n^\eta < n^{\lambda+1}$. Then it is easy to see that $zn^{\lambda+1} \geq z^{1/2}n^\eta + 4\nu_1 n^{\lambda+1}, \forall n \geq 1$ also holds for any $z \geq \tilde{c}_2$. Therefore, we have the following property:

$$\mathbb{P}\left(\tilde{R}_{n,1} \geq zn^{\lambda+1}\right) \leq \frac{\tilde{c}_1^{\xi+1}\tilde{c}_2^{\alpha-3}\beta}{z^{(\alpha-3)/2}}, \ \forall z \geq 1. \tag{12}$$

To see this, first suppose that $z \geq \tilde{c}_2$; then $zn^{\lambda+1} \geq z^{1/2}n^{\eta} + 4\nu_1 n^{\lambda+1}, \forall n \geq 1$ and since $\tilde{c}_2 \geq 1$, we have $\mathbb{P}\left(\tilde{R}_{n,1} \geq zn^{\lambda+1}\right) \leq \mathbb{P}\left(\tilde{R}_{n,1} \geq \frac{z^{1/2}}{\tilde{c}_2}n^{\eta} + 4\nu_1\rho^H n\right) \leq \frac{\tilde{c}_1^{\xi+1}\tilde{c}_2^{\alpha-3}\beta}{z^{(\alpha-3)/2}}$. On the other hand, if $1 \leq z < \tilde{c}_2$, then the inequality (8) trivially holds, because $\tilde{c}_2^{\alpha-3} > z^{\alpha-3} \geq z^{(\alpha-3)/2}$ and $\beta > 1, \tilde{c}_1 \geq 1$, making the RHS greater than 1. The other side of the concentration inequality follows similarly. This completes the analysis for $\tilde{R}_{n,1}$.

Similarly, as for the regret from $\mathcal{L}_3$, we have the following result:

$$\mathbb{P}\left(\tilde{R}_{n,3} \geq zn^{\lambda+1}\right) \leq \frac{\tilde{c}_9^{\alpha-3}(\tilde{c}_8^{\xi+1}\beta + \tilde{c}_7)}{z^{(\alpha-3)/2}}, \ \forall z \geq 1, \tag{13}$$

where again $\tilde{c}_7, \tilde{c}_8, \tilde{c}_9$ are constant independent of $n$ and $z$.

### B.5   Completing proof of concentration

First, recall that the inequalities (8)(10)(11)(12)(13) still hold even when $0 < z < 1$. This is because the RHS of the inequalities will be greater than 1, which is a trivial upper bound for a probability value. Putting together the bounds we got for each individual term, for every $z \geq 1$, we have

$$\mathbb{P}\left(R_n \geq zn^{\lambda+1}\right) \leq \sum_{i=1}^{3}\mathbb{P}\left(R_{n,i} \geq \frac{z}{5}n^{\lambda+1}\right) + \sum_{i=1}^{2}\mathbb{P}\left(\tilde{R}_{n,i} \geq \frac{z}{5}n^{\lambda+1}\right) \leq \frac{\beta'}{z^{(\alpha-3)/2}},$$

where $\beta' > 1$ is a constant independent of $n$ and $z$. Therefore, we have the desired concentration property:

$$\mathbb{P}(\sum_{t=1}^{n} Y_t - nf^* \geq n^{\eta'}z) \leq \frac{\beta'}{z^{\xi'}}, \tag{14}$$

where $\xi' = (\alpha - 3)/2, \eta' = \lambda + 1 = \frac{\frac{\alpha}{\xi(1-\eta)}+d'+\frac{1}{1-\eta}}{1+d'+\frac{1}{1-\eta}}$, and $\beta' > 1$ depends on $\alpha, \beta, \eta, \xi$ and $\bar{H}$. The other side of the concentration inequality follows similarly.

### B.6   Convergence results

We conclude with a convergence analysis of the regret. Let $R_n = \sum_{t=1}^{n}(f^* - Y_t)$ denote the regret of Algorithms 2 and 3 with the depth limitation $\bar{H}$. In the following, we proceed with the special case that there is only one optimal node on depth $\bar{H}$, i.e., there is only one node $(\bar{H}, I^*)$ on depth $\bar{H}$ with $\Delta_{\bar{H},I^*} \leq 2\nu_1\rho^{\bar{H}}$, which in turn implies $\mathcal{P}_{\bar{H},I^*} \subseteq \mathcal{X}_{4\nu_1\rho^{\bar{H}}}$ (Lemma 2). The regret of the general case with multiple optimal nodes is bounded by a constant multiple of this special case.

We partition the regret into three parts, but in a way that is slightly different from the previous concentration analysis. Let $R_n = R_{\mathcal{T}} + R_{n,1} + R_{n,3}$, where $R_{\mathcal{T}}$ denotes the regret above depth $\bar{H}$, $R_{n,1}$ denotes the regret from $\mathcal{L}_1$ (the set of nodes on depth $\bar{H}$ that are descendants of nodes in $I_H$), and $R_{n,3}$ denotes the regret from $\mathcal{L}_3$ (the set of nodes on depth $\bar{H}$ that are descendants of nodes in $\cup_{0 \leq h \leq H}J_h$). Recall that the bandit rewards are bounded in $[-R, R]$. Then it is easy to see that $R_{\mathcal{T}}$ is bounded by a constant, denoted by $C_1$, because the number of nodes played above depth $\bar{H}$ is upper bounded by a constant independent of $n$.

Now we consider $R_{n,1}$. Any node in $I_H$ is by definition $2\nu_1\rho^H$-optimal. By Lemma 2, the domain of $I_H$ lies in $\mathcal{X}_{4\nu_1\rho^H}$, and we know the descendants of $I_H$ also lie in the domain of $\mathcal{X}_{4\nu_1\rho^H}$, satisfying $\sum_{t=1}^{n}(f^* - f(X_t))\,\mathbb{I}_{\{(H_t,I_t)\in\mathcal{L}_1\}} \leq 4\nu_1\rho^H n$. Let $\tilde{n}_1 = |\mathcal{L}_1|$, and let $\mathbb{I}_{\{\cdot\}}$ denote $\mathbb{I}_{\{(H_t,I_t)\in\mathcal{L}_1\}}$ for

short. Then we have

$$\mathbb{E}\left[R_{n,1}\right] = \mathbb{E}\left[\sum_{t=1}^{n}(f^* - Y_t)\mathbb{I}_{\{(H_t,I_t)\in\mathcal{L}_1\}}\right]$$

$$= \mathbb{E}\left[\sum_{t=1}^{n}(f^* - f(X_t))\mathbb{I}_{\{(H_t,I_t)\in\mathcal{L}_1\}}\right] + \mathbb{E}\left[\sum_{t=1}^{n}(f(X_t) - Y_t)\mathbb{I}_{\{(H_t,I_t)\in\mathcal{L}_1\}}\right]$$

$$\leq 4n\nu_1\rho^H + \mathbb{E}\left[\sum_{t=1}^{n}(f(X_t) - f_t(X_t))\mathbb{I}_{\{\cdot\}}\right] + \mathbb{E}\left[\sum_{t=1}^{n}(f_t(X_t) - Y_t)\mathbb{I}_{\{\cdot\}}\right]$$

$$\leq 4n\nu_1\rho^H + \sum_{t=1}^{n}\frac{C}{t^\zeta},$$

where the last step holds due to the definition of the mean-payoff function that $\mathbb{E}\left[Y_t\right] = \mathbb{E}\left[f_t(X_t)\right]$ and the convergence property of $f_t$. Since $\sum_{t=1}^{n}\frac{1}{t^\zeta} \leq \int_0^n t^{-\zeta} \leq \frac{n^{1-\zeta}}{1-\zeta}$, there exists some constant $C_2$ such that

$$\frac{1}{n}\mathbb{E}\left[R_{n,1}\right] \leq \frac{1}{n}\left(4n\nu_1\rho^H + \frac{Cn^{1-\zeta}}{1-\zeta}\right)$$

$$\leq 4\nu_1\rho^H + \frac{C}{(1-\zeta)n^\zeta}$$

$$\leq \frac{C_2}{n^\zeta},$$

where the last step is by the fact that $\rho^H = n^\lambda$ and that $\zeta \leq -\lambda$.

Finally, we analyze the regret of $R_{n,3}$. Let $\tilde{n}_3 = |\mathcal{L}_3|$. For any node $(h,i) \in \mathcal{L}_3$, since the parent of any $(h,i) \in J_h$ is in $I_{h-1}$, we know by Lemma 2 that the domain of $(h,i)$ is in $\mathcal{X}_{4\nu_1\rho^{h-1}}$. Further, $(h,i)$ is not $2\nu_1\rho^h$-optimal by the definition of $J_h$. We then have

$$\mathbb{E}\left[R_{n,3}\right] = \mathbb{E}\left[\sum_{t=1}^{n}(f^* - Y_t)\mathbb{1}_{\{(H_t,I_t)\in\mathcal{L}_3\}}\right]$$

$$= \mathbb{E}\left[\sum_{t=1}^{n}(f^* - f(X_t))\mathbb{1}_{\{(H_t,I_t)\in\mathcal{L}_3\}}\right] + \mathbb{E}\left[\sum_{t=1}^{n}(f(X_t) - Y_t)\mathbb{1}_{\{(H_t,I_t)\in\mathcal{L}_3\}}\right]$$

$$\leq \sum_{h=1}^{H} 4\nu_1\rho^{h-1} \sum_{i:(h,i)\in J_h} \mathbb{E}\left[T_{h,i}(n)\right] + \frac{C}{(1-\zeta)n^{\zeta-1}}$$

$$\leq \sum_{h=1}^{H} 4\nu_1\rho^{h-1}|J_h|\left[\left(\frac{2n^{\alpha/\xi}}{\nu_1\rho^h}\right)^{\frac{1}{1-\eta}} + 2 + \frac{1}{\alpha-3}\right] + \frac{C}{(1-\zeta)n^{\zeta-1}}$$

where the last step is by an application of Lemma 7. Further, since the parent of $J_h$ is in $I_{h-1}$, we know from Lemma 3 that $|J_h| \leq 2|I_{h-1}| \leq 2C_3\left(\nu_2\rho^{h-1}\right)^{-d'}$ for some constant $C_3$. Therefore, there exists some constant $C_4$, such that

$$\frac{1}{n}\mathbb{E}\left[R_{n,3}\right] \leq \frac{1}{n}\sum_{h=1}^{H} 8C_3\nu_1\rho^{h-1}\left(\nu_2\rho^{h-1}\right)^{-d'}\left[\left(\frac{2n^{\alpha/\xi}}{\nu_1\rho^h}\right)^{\frac{1}{1-\eta}} + 2 + \frac{1}{\alpha-3}\right] + \frac{C}{(1-\zeta)n^\zeta} \leq \frac{C_4}{n^\zeta},$$

where the last step holds because $\frac{1}{n}\sum_{h=1}^{H} 8C_3\nu_1\rho^{h-1}\left(\nu_2\rho^{h-1}\right)^{-d'}\left(\frac{2n^{\alpha/\xi}}{\nu_1\rho^h}\right)^{\frac{1}{1-\eta}}$ is in the order of $O(n^\lambda)$, and by the fact that $\zeta \leq -\lambda$.

Putting everything together, we arrive at the desired convergence result:

$$\left|f^* - \frac{1}{n}\mathbb{E}\left[\sum_{t=1}^{n}Y_t\right]\right| = \left|\frac{1}{n}\mathbb{E}\left[R_n\right]\right| = \left|\frac{1}{n}\mathbb{E}\left[R_\mathcal{T} + R_{n,1} + R_{n,3}\right]\right| \leq \frac{C_0}{n^\zeta},$$

where $C_0 > 0$ is a proper constant that can be calculated from $C, R, \alpha, \nu_1, \bar{H}$ and $\zeta$.

# C  Proof of Theorem 1

In the following, we provide a complete proof for Theorem 1. The idea of this proof is built upon the analysis of fixed-depth Monte-Carlo tree search derived in Shah et al. (2019). Given the value function oracle $\hat{V}$ at the leaf nodes, a depth-$D$ MCTS can be approximately considered as $D$ steps of value iteration starting from $\hat{V}$. Let $V^{(d)}$ be the value function after $d$ steps of exact value iteration with $V^{(0)} = \hat{V}$. Since value iteration is a contraction mapping with respect to the $L^{\infty}$ norm, we have $\left\| V^{(d+1)} - V^* \right\|_{\infty} \le \gamma \left\| V^{(d)} - V^* \right\|_{\infty}$, where $V^*$ is the optimal value function. Therefore, we conclude that

$$\left| V^{(D)}(s^{(0)}) - V^*(s^{(0)}) \right| \le \gamma^D \left\| \hat{V} - V^* \right\|_{\infty} = \gamma^D \varepsilon_0 \tag{15}$$

for the MCTS root node $s^{(0)}$.

In the following, we will show that the empirical average reward collected at the root node of MCTS (denoted as $\tilde{v}^{(0)}(s^{(0)})/n$ in Algorithm 1) is within $O(n^{\eta-1})$ of $V^{(D)}(s^{(0)})$ after $n$ rounds of MCTS simulations. The proof is based on an inductive procedure that we will go through in the following sections. Before that, we first introduce a lemma that will be useful throughout.

**Lemma 1.** *Consider real-valued random variables $X_i, Y_i$ for $i \ge 1$, where $X_i$'s are independent and identically distributed, taking values in $[-B, B]$ for some $B > 0$. $Y_i$'s are independent of $X_i$'s, satisfying the following two properties:*

*A. Convergence: Let $\bar{Y}_n = \frac{1}{n} \left( \sum_{i=1}^{n} Y_i \right)$; then there exists $C > 0, 0 < \zeta \le 1/2$, and $\mu_Y$, such that for every integer $n \ge 1$*

$$\left| \mathbb{E}\left[ \bar{Y}_n \right] - \mu_Y \right| \le \frac{C}{n^{\zeta}} \tag{16}$$

*B. Concentration: There exist constants $\beta > 1, \xi > 0$, and $1/2 \le \eta < 1$, such that for every $z \ge 1$ and every integer $n \ge 1$:*

$$\mathbb{P}\left( n\bar{Y}_n - n\mu_Y \ge n^{\eta} z \right) \le \frac{\beta}{z^{\xi}}, \quad \mathbb{P}\left( n\bar{Y}_n - n\mu_Y \le -n^{\eta} z \right) \le \frac{\beta}{z^{\xi}}. \tag{17}$$

*Let $Z_i = X_i + \gamma Y_i$ for some $0 < \gamma < 1$, and let $\bar{Z}_n = \frac{1}{n} \sum_{i=1}^{n} Z_i = \frac{1}{n} \sum_{i=1}^{n} (X_i + \gamma Y_i)$. Define $\mu_X = \mathbb{E}[X_1]$. Then, the following properties are satisfied:*

*A. Convergence:*

$$\left| \mathbb{E}\left[ \bar{Z}_n \right] - (\mu_x + \gamma \mu_Y) \right| \le \frac{C}{n^{\zeta}} \tag{18}$$

*B. Concentration: There exists a constant $\beta' > 1$ depending on $\gamma, \xi, \beta$ and $B$, such that for every $z \ge 1$ and every integer $n \ge 1$:*

$$\mathbb{P}\left( n\bar{Z}_n - n(\mu_X + \gamma \mu_Y) \ge n^{\eta} z \right) \le \frac{\beta'}{z^{\xi}},$$

$$\mathbb{P}\left( n\bar{Z}_n - n(\mu_X + \gamma \mu_Y) \le -n^{\eta} z \right) \le \frac{\beta'}{z^{\xi}}.$$

*Proof.* We first prove the convergence property of $\bar{Z}_n$. $\left| \mathbb{E}\left[ \bar{Z}_n \right] - (\mu_X + \gamma \mu_Y) \right| = \left| \gamma \mathbb{E}\left[ \bar{Y}_n \right] - \gamma \mu_Y \right| \le \frac{\gamma C}{n^{\zeta}} \le \frac{C}{n^{\zeta}}$.

We then prove the concentration property of $\bar{Z}_n$. Let $\bar{X}_n = \frac{1}{n} \sum_{i=1}^{n} X_i$. By Hoeffding's inequality, we know $\mathbb{P}\left( \bar{X}_n - \mu_X \ge \varepsilon \right) \le \exp(\frac{-2n\varepsilon^2}{B^2})$. Then,

$$\begin{aligned}
&\mathbb{P}\left( n\bar{Z}_n - n(\mu_X + \gamma \mu_Y) \ge n^{\eta} z \right) \\
=&\mathbb{P}\left( n\bar{X}_n - n\mu_X + n\gamma \bar{Y}_n - n\gamma \mu_Y \ge n^{\eta} z \right) \\
\le&\mathbb{P}\left( n\bar{X}_n - n\mu_X \ge \frac{n^{\eta} z}{2} \right) + \mathbb{P}\left( n\bar{Y}_n - n\mu_Y \ge \frac{n^{\eta} z}{2\gamma} \right) \\
\le&\exp\left( -\frac{n^{2\eta-1} z^2}{2B^2} \right) + \frac{2^{\xi} \beta \gamma^{\xi}}{z^{\xi}} \\
\le&\frac{\beta'}{z^{\xi}}
\end{aligned}$$

where $\beta'$ is a constant large enough depending on $\gamma, \xi, \beta$ and $B$. The other side of the concentration inequality follows similarly. □

## C.1 Base case

We wanted to inductively show that the empirical mean reward collected at the root node of MCTS is within $O(n^{\eta-1})$ of the value iteration result $V^{(D)}(s^{(0)})$ after $n$ rounds of MCTS simulations. We start with the induction base case at MCTS depth $D-1$, which contains the parent nodes of the leaf nodes at level $D$.

First, notice that there are only finitely many nodes at MCTS depth $D-1$ when $n$ goes to infinity, even though both the state space and the action space are continuous. This is because the HOO tree has limited depth at each MCTS node, and we repeatedly take the same action at a leaf of the HOO tree, resulting in a finite number of actions tried at each state. Further, we have assumed deterministic transitions, and thus each action at a given state repeatedly leads to the same destination state throughout the MCTS process. Combining those two properties gives finite number of nodes in the MCTS tree.

Consider a node denoted as $i$ at depth $D-1$, and let $s_{i,D-1}$ denote the corresponding state. According to the definition of Algorithm 1, whenever state $s_{i,D-1}$ is visited, the bandit algorithm will select an action $a$ from the action space, and the environment will transit to state $s'_D = s_{i,D-1} \circ a$ at depth $D$. The corresponding reward collected at node $i$ of depth $D-1$ would be $R(s_{i,D-1}, a) + \gamma \tilde{v}^{(D)}(s'_D)$, where the reward $R(s, a)$ is an independent random variable taking values bounded in $[-R_{max}, R_{max}]$. Recall that we use a deterministic value function oracle at depth $D$, and hence $\tilde{v}^{(D)}(s'_D) = \hat{V}(s'_D)$ is fully determined once the action $a$ is known. We also know the reward is bounded in $[-\frac{R_{max}}{1-\gamma} - \varepsilon_0, \frac{R_{max}}{1-\gamma} + \varepsilon_0]$, where $\varepsilon_0$ is the largest possible mistake made by the value function oracle. We can then apply Lemma 1 here, with the $X$'s in Lemma 1 corresponding to the partial sums of independent rewards $R(s_{i,D-1}, a)$, the $Y$'s corresponding to the deterministic values $\tilde{v}^{(D)}(s'_D)$. From the result of Lemma 1, we know for the given $\alpha^{(D-1)}, \eta^{(D-1)}$ and $\xi^{(D-1)}$ calculated from (3), there exists a constant $\beta^{(D-1)}$ such that the rewards collected at $s_{i,D-1}$ satisfy the concentration property (5) required by Theorem 2.

Further, let $f_n$ in Theorem 2 be the mean-payoff function when state $s_{i,D-1}$ is visited for the $n$-th time, i.e., $f_n(a) = \mathbb{E}\left[R(s_{i,D-1}, a)\right] + \gamma \hat{V}(s'_D)$. Then since the rewards are stationary, there apparently exists a function $f = f_n, \forall n \geq 1$ such that the convergence (4) property is satisfied with arbitrary value of $\zeta$ such that $0 < \zeta < 1 - \frac{\alpha}{\xi(1-\eta)}$. Since we use exactly the same Algorithms 2 and 3 in the MCTS simulations as the ones stated in Theorem 2, the results of Theorem 2 apply.

Finally, define

$$\mu_*^{(D-1)}(s_{i,D-1}) = \sup_{a \in A} \left\{ \mathbb{E}\left[R(s_{i,D-1}, a)\right] + \gamma \tilde{v}^{(D)}(s_{i,D-1} \circ a) \right\}.$$

Applying Theorem 2 gives the following result:

**Proposition 1.** *Consider a node $i$ at depth $D-1$ of MCTS with the corresponding state $s_{i,D-1}$. Let $\tilde{v}_n^{(D-1)}(s_{i,D-1})$ denote the value of $\tilde{v}^{(D-1)}(s_{i,D-1})$ at the end of the $n$-th round of MCTS simulations. Then, for a given $\xi^{(D-1)} > 0, \eta^{(D-1)} \in [\frac{1}{2}, 1), \alpha^{(D-1)} > 3$, and a proper value of $\beta^{(D-1)}$ given by Lemma 1, we have*

*A. Convergence: There exists some constant $C_0 > 0$ and $0 < \zeta^{(D-1)} < 1 - \frac{\alpha^{(D-1)}}{\xi^{(D-1)}(1-\eta^{(D-1)})}$, such that*

$$\left| \frac{1}{n}\mathbb{E}\left[\tilde{v}_n^{(D-1)}(s_{i,D-1}) - \mu_*^{(D-1)}(s_{i,D-1})\right] \right| \leq \frac{C_0}{n^{\zeta^{(D-1)}}}.$$

*B. Concentration: There exist constants $\beta' > 1, \xi' > 0$, and $1/2 \leq \eta' < 1$, such that for every $z \geq 1$ and every integer $n \geq 1$:*

$$\mathbb{P}\left(\tilde{v}_n^{(D-1)}(s_{i,D-1}) - n\mu_*^{(D-1)}(s_{i,D-1}) \geq n^{\eta'}z\right) \leq \frac{\beta'}{z^{\xi'}},$$

$$\mathbb{P}\left(\tilde{v}_n^{(D-1)}(s_{i,D-1}) - n\mu_*^{(D-1)}(s_{i,D-1}) \leq -n^{\eta'}z\right) \leq \frac{\beta'}{z^{\xi'}},$$

*where* $\eta' = \frac{\frac{\alpha^{(D-1)}}{\xi^{(D-1)}(1-\eta^{(D-1)})}+d'+\frac{1}{1-\eta^{(D-1)}}}{1+d'+\frac{1}{1-\eta^{(D-1)}}}$ *with constant* $d'$ *defined in Definition 3,* $\xi' = (\alpha^{(D-1)} - 3)/2$, *and* $\beta' > 1$ *depends on* $\alpha^{(D-1)}, \beta^{(D-1)}, \eta^{(D-1)}, \xi^{(D-1)}$ *and* $\bar{H}$.

Since $\alpha^{(D-1)} < \xi^{(D-1)}(1-\eta^{(D-1)})$, we can see $0 < \eta' < 1$. We would also like to remark that the definition of $\mu_*^{D-1}(s_{i,D-1})$ is exactly the value function estimation at $s_{i,D-1}$ after one step of value iteration starting from $\hat{V}$. If we set $\alpha^{(D-1)} = \xi^{(D-1)}\eta^{(D-1)}(1-\eta^{(D-1)})$, then $\zeta^{(D-1)} \in (0, \frac{1}{2})$. This completes the base case for our induction.

## C.2 Induction step

We have shown that the convergence and concentration requirements are satisfied from depth $D$ to depth $D-1$. In the following, we will recursively show that these properties also hold from depth $d$ to depth $d-1$ for all $1 \le d \le D-1$.

Consider a node denoted as $i$ at depth $d-1$, and let $s_{i,d-1}$ denote the corresponding state. Again, according to the definition of Algorithm 1, whenever state $s_{i,d-1}$ is visited, the bandit algorithm will select an action $a$ from the action space, and the environment will transit to state $s'_d = s_{i,d-1} \circ a$ at depth $d$. The corresponding reward collected at node $i$ of depth $d-1$ would be $R(s_{i,d-1}, a) + \gamma\tilde{v}^{(d)}(s'_d)$, where the reward $R(s, a)$ is an independent random variable taking values bounded in $[-R_{max}, R_{max}]$. Our induction hypothesis assumes that $\tilde{v}^{(d)}$ satisfies the convergence and concentration properties for all states at depth $d$, with parameters $\alpha^{(d)}, \xi^{(d)}, \eta^{(d)}$ defined by (3) and proper value of $\beta^{(d)}$. Therefore, we can again apply Lemma 1 here, with the $X$'s in Lemma 1 corresponding to the partial sums of independent rewards $R(s_{i,d-1}, a)$, and the $Y$'s corresponding to $\tilde{v}^{(d)}(s'_d)$ that satisfy the convergence and concentration properties by our induction hypothesis. From the result of Lemma 1, we know for the given $\alpha^{(d-1)}, \eta^{(d-1)}$ and $\xi^{(d-1)}$ calculated from (3), there exists a constant $\beta^{(d-1)}$ such that the rewards collected at $s_{i,d-1}$ satisfy the concentration property (5) required by Theorem 2.

Let $f_n$ in Theorem 2 be the mean-payoff function after state $s_{i,D-1}$ is visited for the $n$-th time, i.e., $f_n(a) = \mathbb{E}\left[R(s_{i,D-1}, a)\right] + \gamma\tilde{v}_n^{(d)}(s'_d)/n$. Define $f(a) = \mathbb{E}\left[R(s_{i,D-1}, a)\right] + \gamma\mu_*^{(d)}(s'_d)$, then we can see the convergence requirement (4) is also satisfied by $f_n$ and $f$, with $\zeta = \zeta^{(d)}$. Therefore, the results of Theorem 2 apply.

Finally, define

$$\mu_*^{(d-1)}(s_{i,d-1}) = \sup_{a \in A}\left\{\mathbb{E}\left[R(s_{i,d-1}, a)\right] + \gamma\mu_*^{(d)}(s_{i,d-1} \circ a)\right\}.$$

A direct application of Theorem 2 gives the following result:

**Proposition 2.** *For a node* $i$ *at depth* $d-1$ *of MCTS with the corresponding state* $s_{i,d-1}$. *Let* $\tilde{v}_n^{(d-1)}(s_{i,d-1})$ *denote the value of* $\tilde{v}^{(d-1)}(s_{i,d-1})$ *at the end of the* $n$-th *round of MCTS simulations. Then, for a given* $\xi^{(d-1)} > 0, \eta^{(d-1)} \in [\frac{1}{2}, 1), \alpha^{(d-1)} > 3$, *and a proper value of* $\beta^{(d-1)}$ *given by Lemma 1, we have*

*A. Convergence: There exists some constant* $C_0 > 0$ *and* $0 < \zeta^{(d-1)} < 1 - \frac{\alpha^{(d-1)}}{\xi^{(d-1)}(1-\eta^{(d-1)})}$, *such that*

$$\left|\frac{1}{n}\mathbb{E}\left[\tilde{v}_n^{(d-1)}(s_{i,d-1}) - \mu_*^{(d-1)}(s_{i,d-1})\right]\right| \le \frac{C_0}{n^{\zeta^{(d-1)}}}. \tag{19}$$

*B. Concentration: There exist constants* $\beta' > 1, \xi' > 0$, *and* $1/2 \le \eta' < 1$, *such that for every* $z \ge 1$ *and every integer* $n \ge 1$:

$$\mathbb{P}\left(\tilde{v}_n^{(d-1)}(s_{i,d-1}) - n\mu_*^{(d-1)}(s_{i,d-1}) \ge n^{\eta'}z\right) \le \frac{\beta'}{z^{\xi'}},$$

$$\mathbb{P}\left(\tilde{v}_n^{(d-1)}(s_{i,d-1}) - n\mu_*^{(d-1)}(s_{i,d-1}) \le -n^{\eta'}z\right) \le \frac{\beta'}{z^{\xi'}},$$

*where* $\eta' = \frac{\frac{\alpha^{(d-1)}}{\xi^{(d-1)}(1-\eta^{(d-1)})}+d'+\frac{1}{1-\eta^{(d-1)}}}{1+d'+\frac{1}{1-\eta^{(d-1)}}}$ *with constant* $d'$ *defined in Definition 3,* $\xi' = (\alpha^{(d-1)} - 3)/2$, *and* $\beta' > 1$ *depends on* $\alpha^{(d-1)}, \beta^{(d-1)}, \eta^{(d-1)}, \xi^{(d-1)}$ *and* $\bar{H}$.

Since $\alpha^{(d-1)} < \xi^{(d-1)}(1-\eta^{(d-1)})$, we can see that $0 < \eta' < 1$. If we set $\alpha^{(d-1)} = \xi^{(d-1)}\eta^{(d-1)}(1-\eta^{(d-1)})$, then $\zeta^{(d-1)} \in (0, \frac{1}{2})$. Notice that the definition of $\mu_*^{d-1}(s_{i,d-1})$ is exactly the value function estimation at $s_{i,d-1}$ after $D - d$ steps of value iteration starting from $\hat{V}$. This completes the proof of the induction step.

### C.3  Completing proof of Theorem 1

Following an inductive procedure, we can see that the convergence result (19) also holds at the MCTS root node $s^{(0)}$. After $n$ rounds of MCTS simulations starting from the root node, the empirical mean reward collected at $s^{(0)}$ satisfies:

$$\left| \frac{1}{n} \mathbb{E} \left[ \tilde{v}_n^{(0)}(s^{(0)}) - \mu_*^{(0)}(s^{(0)}) \right] \right| \leq \frac{C_0}{n^{\zeta^{(0)}}}, \tag{20}$$

where $\mu_*^{(0)}(s^{(0)})$ is the value function estimation for $s^{(0)}$ after $D$ rounds of value iteration starting from $\hat{V}$, and $\zeta^{(0)} \in (0, \frac{1}{2})$ if we set $\alpha^{(0)} = \xi^{(0)}\eta^{(0)}(1 - \eta^{(0)})$. Recall from Equation (15) that $\left| \mu_*^{(0)}(s^{(0)}) - V^*(s^{(0)}) \right| \leq \gamma^D \left\| \hat{V} - V^* \right\|_\infty = \gamma^D \varepsilon_0$. By the triangle inequality, we conclude that

$$\left| \frac{1}{n} \mathbb{E} \left[ \tilde{v}_n^{(0)}(s^{(0)}) - V^*(s^{(0)}) \right] \right| \leq O \left( \frac{1}{n^\zeta} \right) + \gamma^D \varepsilon_0,$$

for some $0 < \zeta < 1/2$. This completes the proof of Theorem 1.

## D  Technical Lemmas

**Lemma 2.** *(Lemma 3 in Bubeck et al. (2011)) Under Assumptions 1 and 2, for some region $\mathcal{P}_{h,i}$, if $\Delta_{h,i} \leq c\nu_1 \rho^h$ for some constant $c \geq 0$, then all the arms in $\mathcal{P}_{h,i}$ are $\max\{2c, c+1\}$-optimal.*

*Proof.* This lemma is stated in exactly the same as way Lemma 3 in Bubeck et al. (2011), and we therefore omit the proof here. □

**Lemma 3.** *There exists some constant $C > 0$, such that $|I_h| \leq C(\nu_2 \rho^h)^{-d'}$ for all $h \geq 0$.*

*Proof.* This result is the same as the second step in the proof of Theorem 6 in Bubeck et al. (2011). We therefore omit the proof here. □

**Lemma 4.** *Let Assumptions 1 and 2 hold. Then for every optimal node [3] $(h, i)$ and any integer $n \geq 1$, there exists a constant $\beta_1 > 1$, such that*

$$\mathbb{P} \left( U_{h,i}(n) \leq f^* \right) \leq \frac{\beta_1}{n^{\alpha-1}}.$$

*Proof.* If $(h, i)$ is not played during the first $n$ rounds, then by assumption $U_{h,i}(n) = \infty$ and the inequality trivially holds. Now we focus on the case where $T_{h,i}(n) \geq 1$. From Lemma 2, we know that $f^* - f(x) \leq \nu_1 \rho^h$, $\forall x \in \mathcal{P}_{h,i}$. Then we have $\sum_{t=1}^{n} \left( f(X_t) + \nu_1 \rho^h - f^* \right) \mathbb{I}_{\{(H_t, I_t) \in \mathcal{C}(h,i)\}} \geq 0$.

Therefore,

$$\mathbb{P}\left(U_{h,i}(n) \le f^* \text{ and } T_{h,i}(n) \ge 1\right)$$

$$=\mathbb{P}\left(\widehat{\mu}_{h,i}(n) + n^{\alpha/\xi}T_{h,i}(n)^{\eta-1} + \nu_1\rho^h \le f^* \text{ and } T_{h,i}(n) \ge 1\right)$$

$$=\mathbb{P}\left(T_{h,i}(n)\widehat{\mu}_{h,i}(n) + T_{h,i}(n)\left(\nu_1\rho^h - f^*\right) \le -n^{\alpha/\xi}T_{h,i}(n)^{\eta} \text{ and } T_{h,i}(n) \ge 1\right)$$

$$=\mathbb{P}\left(\sum_{t=1}^{n}\left(Y_t - f\left(X_t\right)\right)\mathbb{I}_{\{(H_t, I_t)\in\mathcal{C}(h,i)\}} + \sum_{t=1}^{n}\left(f\left(X_t\right) + \nu_1\rho^h - f^*\right)\mathbb{I}_{\{(H_t, I_t)\in\mathcal{C}(h,i)\}}\right.$$

$$\left.\le -n^{\alpha/\xi}T_{h,i}(n)^{\eta} \text{ and } T_{h,i}(n) \ge 1\right)$$

$$\le\mathbb{P}\left(\sum_{t=1}^{n}\left(f\left(X_t\right) - Y_t\right)\mathbb{I}_{\{(H_t, I_t)\in\mathcal{C}(h,i)\}} \ge n^{\alpha/\xi}T_{h,i}(n)^{\eta} \text{ and } T_{h,i}(n) \ge 1\right)$$

Since the HOO tree has limited depth, the total number of nodes played in $\mathcal{C}(h,i)$ is upper bounded by some constant $C > 1$ that is independent of $n$. Let $X^j$ denote the $j$-th new node played in $\mathcal{C}(h,i)$, denote the number of times $X^j$ is played as $n_j$, and let $Y_t^j$ ($1 \le t \le n_j$) be the corresponding reward the $t$-th time arm $X^j$ is played. Then, by the union bound, we have

$$\mathbb{P}\left(\sum_{t=1}^{n}\left(f\left(X_t\right) - Y_t\right)\mathbb{I}_{\{(H_t, I_t)\in\mathcal{C}(h,i)\}} \ge n^{\alpha/\xi}T_{h,i}(n)^{\eta} \text{ and } T_{h,i}(n) \ge 1\right)$$

$$\le \sum_{T_{h,i}(n)=1}^{n}\mathbb{P}\left(\sum_{t=1}^{n}\left(f\left(X_t\right) - Y_t\right)\mathbb{I}_{\{(H_t, I_t)\in\mathcal{C}(h,i)\}} \ge n^{\alpha/\xi}T_{h,i}(n)^{\eta}\right)$$

$$= \sum_{T_{h,i}(n)=1}^{n}\mathbb{P}\left(\sum_{j=1}^{\bar{H}}\sum_{t=1}^{n_j}\left(f\left(X^j\right) - Y_t^j\right) \ge n^{\alpha/\xi}T_{h,i}(n)^{\eta}\right)$$

$$\le \sum_{T_{h,i}(n)=1}^{n}\sum_{j=1}^{C}\mathbb{P}\left(\sum_{t=1}^{n_j}\left(f\left(X^j\right) - Y_t^j\right) \ge \frac{n^{\alpha/\xi}}{C}n_j^{\eta}\right)$$

$$\le \frac{\beta_1}{n^{\alpha-1}},$$

where $\beta_1 > 1$ is a constant depending on $C$ and $\beta$, and in the last inequality we applied the concentration property of the bandit problem (5). Notice that we can only use the concentration property when the requirement $z = \frac{n^{\alpha/\xi}}{\bar{H}} \ge 1$ is satisfied, but when $z < 1$, the inequality also trivially holds because $\frac{\beta}{z^\xi} > 1$. This completes the proof of $\mathbb{P}\left(U_{h,i}(n) \le f^*\right) \le \frac{\beta_1}{n^{\alpha-1}}$. $\square$

**Lemma 5.** *(Lemma 14 in Bubeck et al. (2011)) Let* $(h,i)$ *be a suboptimal node. Let* $0 \le k \le h-1$ *be the largest depth such that* $(k, i_k^*)$ *is on the path from the root* $(0,1)$ *to* $(h,i)$, *i.e.,* $(k, i_k^*)$ *is the lowest common ancestor (LCA) of* $(h,i)$ *and the optimal path. Then, for all integers* $u \ge 0$, *we have*

$$\mathbb{E}\left[T_{h,i}(n)\right] \le u + \sum_{t=u+1}^{n}\mathbb{P}\left\{\left[U_{s,i_s^*}(t) \le f^* \text{ for some } s \in \{k+1, \ldots, t-1\}\right]\right.$$

$$\left. \text{or } \left[T_{h,i}(t) > u \text{ and } U_{h,i}(t) > f^*\right]\right\}.$$

*Proof.* This lemma is stated in exactly the same way as Lemma 14 in Bubeck et al. (2011), and the proof follows similarly. We hence omit the proof here. $\square$

**Lemma 6.** *For all integers* $t \le n$, *for any suboptimal node* $(h,i)$ *such that* $\Delta_{h,i} > \nu_1\rho^h$, *and for all integers* $u \ge A_{h,i}(n) = \left\lceil\left(\frac{2n^{\alpha/\xi}}{\Delta_{h,i}-\nu_1\rho^h}\right)^{\frac{1}{1-\eta}}\right\rceil$, *there exists a constant* $\beta_2 > 1$, *such that*

$$\mathbb{P}\left(U_{h,i}(t) > f^* \text{ and } T_{h,i}(t) > u\right) \le \frac{\beta_2 t}{n^{\alpha}}.$$

*Proof.* The proof idea follows almost the same procedure as the proof of Lemma 16 in Bubeck et al. (2011), and we repeat it here due to some minor differences. First, notice that the $u$ defined in the statement of the lemma satisfies $n^{\alpha/\xi}u^{\eta-1} + \nu_1\rho \le \frac{\Delta_{h,i}+\nu_1\rho^h}{2}$. Then we have

$$\mathbb{P}\left(U_{h,i}(t) > f^* \text{ and } T_{h,i}(t) > u\right)$$

$$=\mathbb{P}\left(\widehat{\mu}_{h,i}(t) + n^{\alpha/\xi}u^{\eta-1} + \nu_1\rho^h > f_{h,i}^* + \Delta_{h,i} \text{ and } T_{h,i}(t) > u\right)$$

$$\le\mathbb{P}\left(\widehat{\mu}_{h,i}(t) > f_{h,i}^* + \frac{\Delta_{h,i} - \nu_1\rho^h}{2} \text{ and } T_{h,i}(t) > u\right)$$

$$\le\mathbb{P}\left(T_{h,i}(t)\left(\widehat{\mu}_{h,i}(t) - f_{h,i}^*\right) > \frac{\Delta_{h,i} - \nu_1\rho^h}{2}T_{h,i}(t) \text{ and } T_{h,i}(t) > u\right)$$

$$\le\mathbb{P}\left(\sum_{s=1}^{t}(Y_s - f(X_s))\,\mathbb{I}_{\{(H_s,I_s)\in\mathcal{C}(h,i)\}} > \frac{\Delta_{h,i} - \nu_1\rho^h}{2}T_{h,i}(t) \text{ and } T_{h,i}(t) > u\right)$$

$$\le \sum_{T_{h,i}(t)=u+1}^{t} \mathbb{P}\left(\sum_{s=1}^{t}(Y_s - f(X_s))\,\mathbb{I}_{\{(H_s,I_s)\in\mathcal{C}(h,i)\}} > \frac{\Delta_{h,i} - \nu_1\rho^h}{2}T_{h,i}(t)\right),$$

where in the last step we used the union bound. Then, following a similar procedure as in the proof of Lemma 4 (defining $X^j$ and $Y_t^j$, and then the concentration property), we get:

$$\sum_{T_{h,i}(t)=u+1}^{t} \mathbb{P}\left(\sum_{s=1}^{t}(Y_s - f(X_s))\,\mathbb{I}_{\{(H_s,I_s)\in\mathcal{C}(h,i)\}} > \frac{\Delta_{h,i} - \nu_1\rho^h}{2}T_{h,i}(t)\right)$$

$$\le \sum_{T_{h,i}(t)=u+1}^{t} \frac{\beta_2}{\left(\frac{\Delta_{h,i}-\nu_1\rho}{2}\right)^{\xi}(T_{h,i}(t))^{\xi(1-\eta)}}$$

$$\le \sum_{T_{h,i}(t)=u+1}^{t} \frac{\beta_2}{n^{\alpha}} \le \frac{\beta_2 t}{n^{\alpha}},$$

where $\beta_2 > 1$ is a constant independent of $n$, and in the second step we used the fact that $T_{h,i}(t) > u \ge A_{h,i}(n) = \left\lceil\left(\frac{2n^{\alpha/\xi}}{\Delta_{h,i}-\nu_1\rho^h}\right)^{\frac{1}{1-\eta}}\right\rceil$. This completes our proof of $\mathbb{P}\left(U_{h,i}(t) > f^* \text{ and } T_{h,i}(t) > u\right) \le \frac{\beta_2 t}{n^{\alpha}}$. $\qquad\square$

**Lemma 7.** *For any suboptimal node $(h,i)$ with $\Delta_{h,i} > \nu_1\rho^h$ and any integer $n \ge 1$, there exist constants $\beta_1, \beta_2 > 1$, such that:*

$$\mathbb{E}\left[T_{h,i}(n)\right] \le \left(\frac{2n^{\alpha/\xi}}{\Delta_{h,i} - \nu_1\rho^h}\right)^{\frac{1}{1-\eta}} + 1 + \beta_1 + \frac{\beta_2}{\alpha - 3}.$$

*Proof.* Let $A_{h,i}(n) = \left\lceil\left(\frac{2n^{\alpha/\xi}}{\Delta_{h,i}-\nu_1\rho^h}\right)^{\frac{1}{1-\eta}}\right\rceil$. Then from Lemma 5, we know that

$$\mathbb{E}\left[T_{h,i}(n)\right] \le A_{h,i}(n) + \sum_{t=A_{h,i}(n)+1}^{n}\left(\mathbb{P}\left(T_{h,i}(t) > A_{h,i}(n) \text{ and } U_{h,i}(t) > f^*\right) + \sum_{s=1}^{t-1}\mathbb{P}\left(U_{s,i_s^*}(t) \le f^*\right)\right)$$

By replacing the right hand side with the results from Lemma 4 and Lemma 6, we further have

$$\mathbb{E}\left[T_{h,i}(n)\right] \le A_{h,i}(n) + \sum_{t=A_{h,i}(n)+1}^{n}\left(\frac{\beta_2 t}{n^{\alpha}} + \sum_{s=1}^{t-1}\frac{\beta_1}{t^{\alpha-1}}\right)$$

$$\le A_{h,i}(n) + \frac{\beta_2}{n^{\alpha-2}} + \int_{u}^{n}\frac{\beta_1}{t^{\alpha-2}}dt$$

$$\le \left(\frac{2n^{\alpha/\xi}}{\Delta_{h,i} - \nu_1\rho^h}\right)^{\frac{1}{1-\eta}} + 1 + \beta_2 + \frac{\beta_1}{\alpha - 3}.$$

This completes our proof. □

**Lemma 8.** *Let $(h, i)$ be a suboptimal node. Then for any $n \geq 1$ and any $u > A_{h,i}(n) = \left\lceil \left( \frac{2n^{\alpha/\xi}}{\Delta_{h,i} - \nu_1 \rho^h} \right)^{\frac{1}{1-\eta}} \right\rceil$, there exist constants $\beta_1, \beta_2 > 1$, such that*

$$\mathbb{P}\left( T_{h,i}(n) > u \right) \leq \frac{\beta_2}{n^{\alpha - 2}} + \frac{\beta_1 (u-1)^{3-\alpha}}{\alpha - 3}.$$

*Proof.* Clearly, this inequality holds for $n \leq u$, as $T_{h,i}(n) \leq n$ and the left hand side would be 0 in this case. We therefore focus on the case $n > u$.

We first notice the following monotonicity of the $B$-values: according to the way that $B$-values are defined, the $B$-value of the descendants of a node $(h, i)$ would always be no smaller than the $B$-value of $(h, i)$ itself. Therefore, $B$-values do not decrease along a path from the root to a leaf.

Now, let $0 \leq k \leq h - 1$ be the largest depth such that $(k, i_k^*)$ is on the path from the root $(0, 1)$ to $(h, i)$. We define two events: $E_1 = \{$For each $t \in [u, n]$, $B_{h,i}(t) \leq f^*$ or $T_{h,i}(t) \leq A_{h,i}(t) < u\}$, and $E_2 = \{$For each $t \in [u, n]$, $B_{k+1, i_{k+1}^*}(t) > f^*\}$. We can verify that $E_1 \cap E_2 \subseteq \{T_{h,i}(n) \leq u\}$. To see this, suppose that for some $t \in [u, n]$ we have $B_{h,i}(t) \leq f^*$ and $B_{k+1, i_{k+1}^*}(t) > f^*$; then we know that we would not enter the node $(h, i)$. This is because by the monotonicity of the $B$-values, the ancestor of $(h, i)$ at level $k + 1$ has a $B$-value no larger than $B_{h,i}(t)$, which in turn satisfies $B_{h,i}(t) \leq f^* < B_{k+1, i_{k+1}^*}(t)$. Therefore, we would always enter $B_{k+1, i_{k+1}^*}$ rather than the ancestor of $(h, i)$ at level $k + 1$. In this case, $T_{h,i}$ would not increase at round $t$. Now consider the other case: suppose that for some $t \in [u, n]$ we have $T_{h,i}(t) \leq A_{h,i}(t) < u$ and $B_{k+1, i_{k+1}^*}(t) > f^*$. In this case, we could indeed possibly enter node $(h, i)$ and increase $T_{h,i}$ by 1, but since $T_{h,i}(t) < u$, we still have $T_{h,i}(t+1) \leq u$ after increasing by 1. Considering these two cases inductively, we can see that if $E_1 \cap E_2$ holds, then $T_{h,i}(u-1) < u$ implies $T_{h,i}(n) \leq u$. Since $T_{h,i}(u-1) < u$ trivially holds, we can conclude that $E_1 \cap E_2 \subseteq \{T_{h,i}(n) \leq u\}$.

After we have $E_1 \cap E_2 \subseteq \{T_{h,i}(n) \leq u\}$, we know that $\{T_{h,i}(n) > u\} \subseteq E_1^c \cup E_2^c$, where $E^c$ denotes the complement of event $E$. This in turn gives us $\mathbb{P}\left( \{T_{h,i}(n) > u\} \right) \leq \mathbb{P}\left( E_1^c \right) + \mathbb{P}\left( E_2^c \right)$. From the definition of the $B$-values, $\left\{ B_{k+1, i_{k+1}^*}(t) \leq f^* \right\} \subset \left\{ U_{k+1, i_{k+1}^*}(t) \leq f^* \right\} \cup \left\{ B_{k+2, i_{k+2}^*}(t) \leq f^* \right\}$, and this can be applied recursively up to depth $t$, where the nodes in depth $t$ have not been played at round $t$ and satisfy $B_{t, i_t^*} = \infty > f^*$. Together with the fact that $U_{h,i}(t) \geq B_{h,i}(t)$ (by definition), we have

$$\mathbb{P}\left( T_{h,i}(n) > u \right)$$

$$\leq \mathbb{P}\left( \exists t \in [u, n], B_{h,i}(t) > f^* \text{ and } T_{h,i}(t) > A_{h,i}(t) \right) + \mathbb{P}\left( \exists t \in [u, n], B_{k+1, i_{k+1}^*}(t) \leq f^* \right)$$

$$\leq \mathbb{P}\left( \exists t \in [u, n], U_{h,i}(t) > f^* \text{ and } T_{h,i}(t) > A_{h,i}(t) \right)$$

$$+ \mathbb{P}\left( \exists t \in [u, n], U_{k+1, i_{k+1}^*}(t) \leq f^* \text{ or } U_{k+2, i_{k+2}^*}(t) \leq f^* \text{ or } \dots \text{ or } U_{t-1, i_{t-1}^*}(t) \leq f^* \right)$$

$$\leq \sum_{t=u}^{n} \mathbb{P}\left( U_{h,i}(t) > f^* \text{ and } T_{h,i}(t) > A_{h,i}(t) \right)$$

$$+ \sum_{t=u}^{n} \mathbb{P}\left( U_{k+1, i_{k+1}^*}(t) \leq f^* \text{ or } U_{k+2, i_{k+2}^*}(t) \leq f^* \text{ or } \dots \text{ or } U_{t-1, i_{t-1}^*}(t) \leq f^* \right)$$

$$\leq \sum_{t=u}^{n} \mathbb{P}\left( U_{h,i}(t) > f^* \text{ and } T_{h,i}(t) > A_{h,i}(t) \right) + \sum_{t=u}^{n} \sum_{s=1}^{t-1} \mathbb{P}\left( U_{s, i_s^*}(t) \leq f^* \right),$$

where in the last two steps we used the union bound. Since we know $\mathbb{P}\left(U_{s,i_s^*}(t) \leq f^*\right) \leq \frac{\beta_1}{n^{\alpha-1}}$ from Lemma 4, and $\mathbb{P}\left(U_{h,i}(t) > f^* \text{ and } T_{h,i}(t) > A_{h,i}(t)\right) \leq \frac{\beta_2 t}{n^\alpha}$ from Lemma 6, we conclude that

$$\sum_{t=u}^{n} \mathbb{P}\left(U_{h,i}(t) > f^* \text{ and } T_{h,i}(t) > A_{h,i}(t)\right) + \sum_{t=u}^{n}\sum_{s=1}^{t-1} \mathbb{P}\left(U_{s,i_s^*}(t) \leq f^*\right)$$

$$\leq \sum_{t=u}^{n} \frac{\beta_2 t}{n^\alpha} + \sum_{t=u}^{n}\sum_{s=1}^{t-1} \frac{\beta_1}{t^{\alpha-1}} \leq \sum_{t=u}^{n} \frac{\beta_2 n}{n^\alpha} + \beta_1 \int_{u-1}^{\infty} t^{2-\alpha} dt$$

$$\leq \frac{\beta_2}{n^{\alpha-2}} + \frac{\beta_1 (u-1)^{3-\alpha}}{\alpha - 3}.$$

This completes the proof.

We further remark that if $1 < u \leq n$, then $\frac{1}{n^{\alpha-2}} \leq \frac{u^{3-\alpha}n^{\alpha-3}}{n^{\alpha-2}} \leq \frac{(u-1)^{3-\alpha}}{n}$, which implies

$$\mathbb{P}\left(T_{h,i}(n) > u\right) \leq \frac{\beta_2(u-1)^{3-\alpha}}{n} + \frac{\beta_1(u-1)^{3-\alpha}}{\alpha - 3}. \tag{21}$$

Notice that this inequality also holds when $u > n$, because $T_{h,i}(n) \leq n < u$, and any non-negative value on the RHS is a trivial upper bound for $\mathbb{P}\left(T_{h,i}(n) > u\right)$. $\qquad\square$

*Remark* 4. As a final remark, when we refer to the results of Lemmas 4, 5, 6, 7 and 8, we typically drop the constant factors $\beta_1$ and $\beta_2$ and proceed with $\beta_1 = \beta_2 = 1$ instead. This does not affect our main results up to a constant factor.

## E   Details of the Simulations

In this section, we discuss details of the simulations and empirically evaluate the performance of POLY-HOOT on several classic control tasks. We have chosen three benchmark tasks from the OpenAI Gym (OpenAI, 2016), and extended them to the continuous-action settings as necessary. These tasks include CartPole, Inverted Pendulum Swing-up, and LunarLander.

In the CartPole problem, a pole is attached to a cart through a joint. The task is to apply an appropriate horizontal force to the cart to prevent the pole from falling. For every time step that the pole remains standing (up to 15 degrees from being vertical), a unit reward is given. We have also modified the CartPole problem to a more challenging setting with an increased gravity value (CartPole-IG) to better demonstrate the differences between the algorithms we compare. This new setting requires smoother actions, and bang-bang control strategies easily lead the pole to fall due to the increased momentum. The Inverted Pendulum Swing-up task is also a classic problem in control. A pendulum is attached to a frictionless pivot, starting from a random position. The task is to apply a force to the pendulum to swing it up and let it stay upright. At each time step, a reward is given based on the angle of the current position of the pendulum from being upright. In the LunarLander problem, the task is to design the control signals for a lunar lander to land smoothly on a landing pad. A negative reward is given every time the engine is fired, and a positive reward is given when the lander safely reaches the landing pad.

In the original problem of CartPole, the action set is a discrete set $\{-1, 1\}$. In our CartPole and CartPole-IG environments though, we have extended the action space to a continuous domain $[-1, 1]$. In CartPole-IG, we have further increased the gravity value from $9.8$ to $50$, increased the mass of the pole from $0.1$ to $0.5$, and increased the length of the pole from $1$ to $2$. The other parameters have remained the same as the discrete setting in OpenAI Gym. For the task of Inverted Pendulum, we have manually reduced the randomness of the initial state to ensure that each run of the simulation is initialized more consistently. The reward discount factor was set to be $\gamma = 0.99$ for all the four tasks. The length of the horizon was taken as $T = 150$.

We compare the empirical performance of POLY-HOOT with three continuous MCTS algorithms, including UCT (Kocsis and Szepesvári, 2006) with manually discretized actions, Polynomial Upper Confidence Trees (PUCT) with progressive widening (Auger et al., 2013), and the original empirical implementation of HOOT (Mansley et al., 2011) with a logarithmic bonus term. For all four algorithms, we have set the MCTS depth to be $D = 50$, except for the task of LunarLander where we

Figure 1: Figures (a) and (b) show the rewards of the four algorithms with respect to the rounds of simulations per MCTS step on CartPole and CartPole-IG, respectively. The horizontal axes are in logarithmic scales. The shaded areas denote the standard deviations. Figure (c) shows the reward of discretized-UCT with respect to the action discretization level on CartPole-IG.

set $D = 100$ because this task takes a longer time to finish. We have set the number of simulations at each state to be $n = 100$ rounds. For the UCT algorithm with discretized actions, we have fixed the number of actions to be $10$ and sampled the actions using a uniform grid. For PUCT with progressive widening, we have set the progressive widening coefficient to be $0.5$, i.e., the number of discrete action samples grows at a square-root order in time. For HOOT and `POLY-HOOT`, given the dimension $m$ of the action space, we have calculated the $\rho$ and $\nu_1$ parameters by $\rho = \frac{1}{4^m}$ and $\nu_1 = 4m$. For `POLY-HOOT`, we have set the maximum depth of the HOO tree covering to be $\tilde{H} = 10$, and we have fixed $\alpha = 5, \xi = 20$, and $\eta = 0.5$. The value function oracle we have used is $\hat{V}(s) = 0, \forall s \in S$ for all four algorithms.

In addition to the evaluation results presented in the main text, we have also tested how the number of simulation rounds per planning step influences the rewards of the four algorithms. The number of simulation rounds is proportional to the number of samples used in each step, and hence we can use this experiment to infer the sample complexities of different algorithms. The evaluation results on CartPole and CartPole-IG are shown in Figures 1 (a) and (b), respectively. As we can see, HOOT and `POLY-HOOT` require significantly fewer rounds of simulations to achieve the optimal rewards, which suggests that they have better sample complexities than discretized-UCT and PUCT.

We have also evaluated how the action discretization level influences the performance of discretized-UCT. The evaluation results on CartPole-IG are shown in Figure 1 (c), where different curves denote different numbers of simulation rounds per planning step. As we can see, the performance of discretized-UCT does not necessarily improve with finer granularity of actions. We believe the reason is that, given the fixed number of samples used in each step, each discretized action cannot be well estimated and fully exploited when the discretized action space is large. In addition, there exist huge reward fluctuations even if we only slightly modify the action granularity. This suggests that the performance of discretized-UCT is very sensitive to the discretization level, making this hyper-parameter hard to tune. These evaluation results can further demonstrate the advantages of partitioning the action space adaptively in HOOT and `POLY-HOOT`.

## Footnotes

[3] Recall Definition 4.