[Reviews · NeurIPS 2020]

Review 1

Summary and Contributions: This paper studies MCTS for continuous action settings. Typically MCTS is just useful for discrete action settings and this paper studies the extension to continuous actions with the aim of theoretically justifying the approach taken. The approach is relevant to people interested in planning or people interested in continuous action control (e.g., robotics). The paper first extends an existing UCB-like algorithm for continuous-armed bandits, HOO, by using a polynomial exploration bonus instead of a logarithmic one. This approach is justified by a similar approach in the influential AlphaGo paper and prior work that justifies the approach theoretically for non-stationiary bandit problems. The paper then integrates this enhanced HOO into MCTS and calls the resulting algorithm Poly-HOOT. Theoretical results are given for convergence of approach to optimal action and empirical results show the method out-performs baselines. Overall, I liked the paper and think it clears the acceptance bar.

Strengths: As far as I can tell the approach is novel and I also like that the approach is grounded with theoretical analysis. Empirical evaluation on four toy domains is sound and shows Poly-HOOT performs as well or better than baselines. The performance gain is marginal on most tasks except for LunarLander. However, I think one of the main strengths of this paper is it introduces a theoretically well grounded method while simultaneously improving empirical performance. The paper is of high relevance to the NeurIPS community particularly people working in RL.

Weaknesses: Some empirical comparison is made to prior work (e.g., PUCT and discretized UCT) though other existing methods (discussed in related work) are not compared with. It's unclear why the baselines chosen were chosen. The proposed approach is based on discretization and so it may require many roll-out in higher dimensional problems. It would be nice for the authors to commment on this limitation and how it might compare to other related works.

Correctness: I have not checked all steps of the proofs but believe they're sound. The empirical methodology is correct and reproducible. It would be interesting to add some discussion of wall clock time and time per decision in the experiments.

Clarity: The paper is very well written.

Relation to Prior Work: Yes, the authors include a related works section and cite other approaches for extending MCTS to continuous actions.

Reproducibility: No

Additional Feedback: I read the author's response. Could you comment on how baselines chosen for experiments? Could you comment on the time per decision of methods in experiments? Is it feasible to run Poly-HOOT in a continuous control application where many decisions must be made per second? HOO / HOOT / Poly-HOOT are all based on a method of discretizing the action space. Could you discuss the limitations of this for higher dimensional problems?


Review 2

Summary and Contributions: This paper proposes a Monte-Carlo Tree Search (MCTS) method for continuous action domains by extending Hierarchical Optimistic Optimization (HOO). The proposed method, Poly-HOOT, uses a polynomial term rather than a logarithmic term as the bonus term (bias term) in the UCB1-like formula. Poly-HOOT is proved to converge under some assumptions, such as the bounded depth.

Strengths: This paper provides a Monte-Carlo Tree Search method with proof of convergence and an analysis of the convergence rate, based on a polynomial term. An intuitive question of the researchers in this domain was that how to replace cumulative regret based formula (namely UCB1) with a simple regret based one. One example is the following paper, which uses a polynomial term based regret. J.Y. Audibert, S. Bubeck and R. Munos, Best Arm Identification in Multi-Armed Bandits, COLT-2010. Also, it was empirically known in the computer Go community that using a polynomial term rather than logarithmic term improves the strength. However, there was no tree search algorithm (based on a polynomial term (edited)) which has a proof of convergence. Therefore, the contribution of this paper is important.

Weaknesses: However, there are mainly two concerns. One is the depth bound. I am not entirely convinced that assuming maximum depth is meaningful. The depth bound is not only necessary for the proof but also a hyper-parameter. Another is the experimental settings. Table 1 shows that Poly-HOOT outperformed the counterparts in LunarLander (LunarLanderContinuous-v2). However, the score is averaged over only ten runs, and the average (and the variance or max/min.) is not shown.

Correctness: I couldn't find problems in the method or proof, though I am not confident about the details.

Clarity: In general, it is well written. However, it was difficult to follow the explanation and the proof steps because there was little information that helps the readers. Some examples are, - the intuitive meaning of \alpha, \xi, and \eta is not explained. - I think $n$ means round throughout this paper, which is defined at line 146. However, when $V_n$ or $f_n$ appeared in the equations, I couldn't remember what $n$ meant. It will help the readers if there is an explanation that reminds the readers about the definition and the purpose of the variables.

Relation to Prior Work: This paper provides adequate amount of related work and discusses the novelty of the paper. It focuses on the comparison with HOO and lacks the comparison with simple regret MAB using polynomial term, but in my idea, that is not crucial.

Reproducibility: Yes

Additional Feedback: It would have been more convincing if there had been a discussion of how to determine the parameters, especially the maximum depth, before solving problems. (comment after rebuttal) The feedback adequately answers the questions posed by the reviewers. The comparison with VOOT and additional analysis (including $\bar{H}$) would be valuable.


Review 3

Summary and Contributions: The paper presents an MTCS variant for continuous domains. in each node of the search tree, the continuous action space is explored by a modified variant of the continuous bandit algorithm HOO. In HOO the logarithmic bonus term is replaced by a polynomial one. The main result of the paper is the convergence proof of the POLY-HOOT algorithm. A few empirical evaluation provide further indication of the effectiveness of the algorithm.

Strengths: The convergence proof.

Weaknesses: The experiments could have used more baselines.

Correctness: The paper seems sound.

Clarity: The paper is well written.

Relation to Prior Work: Related literature is discussed.

Reproducibility: Yes

Additional Feedback: POLY-HOOT seems a natural algorithm. MCTS is well established, HOO has been around for a while, and polynomial bonus terms are also popular. Therefore the main contributions is the convergence proof. The algorithm is defined in continuous state space, but given that the transition is assumed deterministic, I do not think that it has any relevance at all (compared to a discrete, but large enough state space). In general, I have reservations about the practicality of MCTS in continuous state/action spaces without a function approximator. VOOT of [Kim et al, 2020] seems to report reasonable practical performance, even compared to some RL algorithms. It would have been useful to include at least VOOT as baseline, and even better some RL methods with function approximation. ----------------------------------- I have read the autors' feedback. I hope that there will be enough time to include some of the additional baseline suggested.


Review 4

Summary and Contributions: This work introduces Polynomial Hierarchical Optimistic Optimization (POLY-HOOT), a Monte0Carlo planning algorithm for solving continuous space MDPs. POLY-HOOT constitutes a modification of the Hierarchical Optimistic Optimization applied to Trees (HOOT) algorithm that integrates MCTS with continuous action bandit algorithm strategy HOO. To be more specific, an HOO is applied at each state and depth in the rollout tree. In contrast to HOOT that uses a logarithmic bonus term, POLY-HOOT uses a polynomial term for bandit exploration. Theoretical results show the POLY-HOOT converges to an arbitrarily small neighborhood of the optimum at a polynomial rate. Experiments have been conducted on four classical rl environments.

Strengths: The main novelty of this work is the replacement of the logarithmic bonus term that used by HOOT algorithm for bandit exploration with a polynomial term. In contrast to HOOT, a theoretical analysis is also provided that proves the non-asymptotic convergence rate of POLY-HOOT algorithm. Moreover, empirical analysis has been conducted on three classical control tasks (CartPole, Inverted pendulum swing up, and LunarLander), showing that POLY-HOOT achieves the same or slightly better performance compared to HOOT. The source code of the experiments is also provided. Finally it should be mentioned

Weaknesses: The general concept of this work is not quite novel. Actually, the proposed POLY-HOOT algorithm is a slightly modification of the standard HOOT. In addition to that I have found the empirical analysis of this work limited. To be more precise, experiments have been conducted only on three classical rl tasks. It could be interesting to check the performance of POLY-HOOT algorithm on some continuous control tasks (Mujoco, roboschool). Additionally, the computational complexity of the POLY-HOOT algorithm should be given.

Correctness: In general, the method and authors claims are valid. Additionally, the empirical evaluation seems to be correct and the source code is provided by the authors.

Clarity: The paper is fairly written and the general idea of the work is presented in a clear way. The only part that should be discussed is the presentation of the empirical results. Also, the authors' justification about the fact that POLY-HOOT outperforms HOOT on the LunarLander environment is not clear. I think that authors should elaborate more about this. Also, authors should describe how they are setting the maximum HOO depth and how does it affect the performance of the POLY-HOOT.

Relation to Prior Work: Generally speaking, the related work and the differences between POLY-HOOT and HOOT are presented adequately by the authors

Reproducibility: Yes

Additional Feedback: Please check my comments on the previous sections. I would like to thank the authors for addressing most of my review points. Therefore, I am going to change my original evaluation by increasing the score.

[Author Response · NeurIPS 2020]

We thank all four reviewers for the constructive feedback. Our main objective was the development of theoretical and
analytical results. Given the reviewers' interest in the empirical performance aspect of our results, we further discuss
the experimental settings below, and provide new and larger-scale experimental results in view of the comments.

**Theoretical significance.** Our main contribution is more on the theoretical/analytic aspect rather than being em-
pirical/algorithmic, serving as a theoretical justification of the integration of MCTS with continuous-armed bandits
(CABs) for continuous MDPs. This intuitive idea has been comprehensively evaluated, with demonstration of excellent
empirical performance. However, rigorous justification for such an approach is missing. Our primary intention was
not to provide more empirical results to support this line of success. Accordingly, we hope that our paper could be
(re)-evaluated with that in mind, as it constitutes an original attempt towards theoretically understanding MCTS for
continuous MDPs. Our theoretical findings have advocated the use of polynomial bonuses, which was largely ignored in
existing empirical solutions; we expect our proof techniques to inspire and guide the design of better empirical methods
as well. Additionally, our Thm 2 constitutes an initial investigation of non-stationary bandits in continuous space—an
important research area, where our techniques might be useful for future developments.

**Detailed Responses**

1. *Novelty of the general concept* (Rev. 4). Our work is far more than an algorithmic "slight modification of the
standard HOOT"; the main contribution/novelty is actually on analysis. First, HOOT is a purely empirical work
and establishing convergence guarantee of HOOT is challenging. We are to offer a first theoretical milestone, and
hopefully, the abstractions/developments here might pave the way for useful finite-time results under stronger structural
assumptions. Additionally, our analytic framework has novel contributions itself. We introduce a new framework to
handle non-stationarity in CABs (by translating $L^\infty$ concentration of non-stationarity into a regret convergence rate),
which is the first in the literature.

2. *Choice of baselines* (Revs. 1 & 3). The 3 baselines we choose, together with (Kim et al., 2020), are the only solutions
we are aware of to MC planning in general continuous spaces. Other methods in *Related Work* either only work in
the discrete setting or require specific/different structures of the problem. We did not include (Kim et al., 2020) in
experiments because it was posted online only very recently, and there was not enough time for us to implement and
evaluate their method. It is indeed a very relevant baseline, and we will include a comparison in the revised version.

3. *Performance improvement seems marginal* (Rev. 1). Our performance gain (over UCT and PUCT) is not marginal
on CartPole-IG and Pendulum. For instance, in CartPole-IG, the pole falls roughly after 120 steps for UCT or PUCT,
while for POLY-HOOT it is steady for at least 2000 steps. The reason that the performance gain "seems" marginal is that
the horizon is set to be 150 (see Appendix F), which is large enough to depict the numerical difference between the
algorithms, but puts POLY-HOOT in an unfavorable situation with "seemingly" insignificant numerical improvement.

4. *Time complexity / Time per decision* (Revs. 1 & 4). It suffices to compare the time complexity (TC) of the bandit
algorithms, as the structures of the search tree are essentially the same. TC for discretized-UCT is $O(KT)$, where $T$ is
the length of the planning horizon and $K$ is the number of discretized actions. For HOOT, it is $O(T \log T)$ as pointed
out by Bubeck et al. (2011). For POLY-HOOT, TC reduces to $\min\{O(\bar{H}T), O(T \log T)\}$ due to the existence of the
depth limitation $\bar{H}$. For PUCT, a loose upper bound is $O(T^{1+\alpha_d})$, where $\alpha_d$ is the progressive widening coefficient,
but in practice the complexity is much lower. We also provide the following "time per decision" results (averaged over
10 runs on a laptop with an Intel Core i5-9300H CPU) on the task of CartPole-IG, with default parameter values.

| Algorithm / $H$ value for POLY-HOOT | discretized-UCT | PUCT | HOOT | 2 | 4 | 6 | 8 | 10 |
|---|---|---|---|---|---|---|---|---|
| Reward | 69.03 | 70.79 | 77.85 | 42.45 | 48.54 | 63.27 | 77.85 | 77.85 |
| Time per decision (s) | 0.950 | 0.305 | 1.173 | 0.054 | 0.149 | 0.610 | 1.030 | 1.057 |

5. *Maximum depth* (Revs. 2 & 4). The maximum depth $\bar{H}$ is indeed important for convergence guarantees. The effect
of $\bar{H}$ should be clear when combined with Lines 19, 24, and 25 in Algorithm 2, where we associate an action with
each node on depth $\bar{H}$. Otherwise, HOO will be allowed to explore a new action at each step, which preempts any
concentration result (as it is not a martingale). In practice, $\bar{H}$ can be viewed as a hyper-parameter to trade off optimality
v.s. computation time. In the above table, we show how $\bar{H}$ influences this trade-off in POLY-HOOT on CartPole-IG. The
computation time of $\bar{H} = 8$ and 10 are close because most of the time POLY-HOOT does not reach the maximum depth
when $\bar{H}$ is large enough. The choice of $\bar{H}$ should depend on the horizon $T$, and this will be detailed in the revision.

6. *Variance of numerical evaluations* (Rev. 2). We repeated our evaluations over 40 runs (previously it was 10), and the
results turned out to be very similar. We do not include the larger-scale results and variances due to space limitations.

7. *Function approximators* (Rev. 3). Function approximators are indeed very important for MCTS to achieve good
empirical performance, especially in continuous spaces. Our goal here is to develop a general analysis for MCTS itself
(not restricted to specific designs). As a theory-oriented paper, it was not our primary intention to optimize the empirical
performance, although we believe that a combination with function approximators will achieve promising performance.

8. *Discussion on LunarLander* (Rev. 4). POLY-HOOT outperforms HOOT more significantly on LunarLander mostly
because of the reward structure of the task itself. Sparse and large rewards cause more severe "non-stationarities", and
HOOT might get trapped to an area of suboptimal actions in the earlier stages. We will provide the variation trajectories
of values inside critical nodes to demonstrate this phenomenon in the final version.

[Meta-Review · NeurIPS 2020]

Reviewers all agreed that this paper presents a novel work towards MCTS in continuous action spaces, with its theoretical analysis making an important contribution.